# Centromere deletion in *Cryptococcus deuterogattii* leads to neocentromere formation and chromosome fusions

**Klaas Schotanus, Joseph Heitman\***

Department of Molecular Genetics and Microbiology, Duke University Medical Center, Durham, United States

**Abstract** The human fungal pathogen *Cryptococcus deuterogattii* is RNAi-deficient and lacks active transposons in its genome. *C. deuterogattii* has regional centromeres that contain only transposon relics. To investigate the impact of centromere loss on the *C. deuterogattii* genome, either centromere 9 or 10 was deleted. Deletion of either centromere resulted in neocentromere formation and interestingly, the genes covered by these neocentromeres maintained wild-type expression levels. In contrast to *cen9Δ* mutants, *cen10Δ* mutant strains exhibited growth defects and were aneuploid for chromosome 10. At an elevated growth temperature (37°C), the *cen10Δ* chromosome was found to have undergone fusion with another native chromosome in some isolates and this fusion restored wild-type growth. Following chromosomal fusion, the neocentromere was inactivated, and the native centromere of the fused chromosome served as the active centromere. The neocentromere formation and chromosomal fusion events observed in this study in *C. deuterogattii* may be similar to events that triggered genomic changes within the *Cryptococcus/Kwoniella* species complex and may contribute to speciation throughout the eukaryotic domain.

**\*For correspondence:**
heitm001@duke.edu

**Competing interests:** The authors declare that no competing interests exist.

## Introduction

Eukaryotic organisms have linear chromosomes with specialized regions: telomeres that cap the ends, origins of replication, and centromeres that are critical for chromosome segregation. During cell division, the centromere binds to a specialized protein complex known as the kinetochore (*Cheeseman, 2014*). Most centromeres are regional, sequence-independent, and defined by the replacement of the canonical histone H3 by the histone homolog CENP-A (CenH3 or Cse4) (*Henikoff and Furuyama, 2010*). In humans, centromeres contain higher-order α-satellite DNA arrays that span 0.1 to 4.8 Mb (*McNulty and Sullivan, 2018*), which is in contrast to most fungal centromeres, which contain transposable elements and repetitive sequences (*Friedman and Freitag, 2017*). Fungal regional centromeres range from the small centromeres of *Candida albicans*, (the CENP-A enriched regions range from 2 to 3.3 kb and are located in 4 to 18 kb open-reading frame ORF-free regions), to the large regional centromeres described in *Neurospora crassa*, (which range from 174 to 287 kb and consist mainly of truncated transposable elements) (*Sanyal et al., 2004*; *Smith et al., 2011*). Similar to mice, some fungi have pericentric regions (*Guenatri et al., 2004*). The most prominent examples are the centromeres of *Schizosaccharomyces pombe*, which have a CENP-A-enriched region comprised of a central core flanked by heterochromatic pericentric regions divided into outer and inner repeats (*Ishii et al., 2008*; *Rhind et al., 2011*). Budding yeast have sequence-dependent centromeres, which are short and have a conserved organization with three centromere DNA elements (consensus DNA elements (CDEs) I-III) (*Kobayashi et al., 2015*). However, the budding yeast *Naumovozyma castellii* has unique consensus DNA elements that differ from those of other budding yeast species (*Kobayashi et al., 2015*).

Infrequently, centromeres can be spontaneously inactivated, resulting in neocentromere formation (i.e., evolutionary new centromeres) (*Ventura et al., 2007*). Neocentromere formation can occur either while the native centromeric sequence is still present on the chromosome or when the native centromere has been mutated or deleted (e.g., from chromosomal rearrangements or γ irradiation damage *Burrack and Berman, 2012*; *Tolomeo et al., 2017*; *Ventura et al., 2007*). In addition, several studies have described neocentromere formation after deletion of native centromeres by molecular genetic engineering in fungi, chickens, and *Drosophila* (*Alkan et al., 2007*; *Ishii et al., 2008*; *Ketel et al., 2009*; *Shang et al., 2013*). In some organisms, the formation of neocentromeres can be deleterious, leading to disease, cancer, or infertility (*Burrack and Berman, 2012*; *Garsed et al., 2014*; *Nergadze et al., 2018*; *Scott and Sullivan, 2014*; *Warburton, 2004*). For example, human neocentromeres are often identified in liposarcomas (*Garsed et al., 2014*). However, neocentromere formation also can be beneficial, leading to speciation (*Ventura et al., 2007*).

Fungal neocentromeres are well described in the diploid yeast *C. albicans* and the haploid fission yeast *S. pombe* (*Ishii et al., 2008*; *Ketel et al., 2009*; *Thakur and Sanyal, 2013*). Deletion of *C. albicans* native centromere 5 or 7 has been shown to induce neocentromere formation and does not result in chromosome loss (*Ketel et al., 2009*; *Thakur and Sanyal, 2013*). In these cases, neocentromeres conferred chromosomal stability similar to the native centromere (*Ketel et al., 2009*; *Mishra et al., 2007*). Deletion of a native centromere in *S. pombe* led to either neocentromere formation or chromosome fusion (*Ishii et al., 2008*; *Ohno et al., 2016*). *S. pombe* neocentromeres formed in telomere-proximal regions near heterochromatin, and neocentromere organization featured a CENP-A-enriched core domain and heterochromatin at the subtelomeric (distal) side. Interestingly, neocentromere formation occurred at the same regions in both wild-type and heterochromatin-deficient strains, suggesting that heterochromatin is dispensable for neocentromere formation in *S. pombe*, although the rate of survival by chromosome fusion was significantly increased in heterochromatin-deficient mutants (*Ishii et al., 2008*). Deletion of kinetochore proteins (*mhf1Δ* and *mhf2Δ*) led to a shift of CENP-A binding, resulting in a CENP-A-enriched region directly adjacent to the native centromere (*Lu and He, 2019*).

In some cases, neocentromeres span genes that are silenced, such as the neocentromeres in *C. albicans*. However the mechanisms that mediate silencing of neocentromeric genes are unknown in *C. albicans,* as proteins that are necessary for heterochromatin formation and gene silencing in other species ( HP1, Clr4, and DNA methyltransferase) are absent in *C. albicans* (*Ketel et al., 2009*). Neocentromeres of *S. pombe* can also span genes. These genes are upregulated during nitrogen starvation and expressed at low levels during stationary growth in wild-type cells, but are silenced under all conditions tested when spanned by neocentromeres. In addition to neocentromeric genes, genes located within native centromeres have been identified in other fungi as well as rice and chicken (*Nagaki et al., 2004*; *Schotanus et al., 2015*; *Shang et al., 2013*).

Recently, the centromeres of the human pathogenic fungus *Cryptococcus deuterogattii* were characterized and compared to those of the closely related species *Cryptococcus neoformans* (centromeres ranging from 27 to 64 kb), revealing dramatically smaller centromeres in *C. deuterogattii* (ranging from 8.7 to 21 kb) (*Janbon et al., 2014*; *Yadav et al., 2018*). *C. deuterogattii* is responsible for an ongoing cryptococcosis outbreak in the Pacific Northwest regions of Canada and the United States (*Fraser et al., 2005*). In contrast to the sister species *C. neoformans*, *C. deuterogattii* commonly infects immunocompetent patients (*Fraser et al., 2005*). *C. deuterogattii* is a haploid basidiomycetous fungus with 14 chromosomes (*D'Souza et al., 2011*; *Farrer et al., 2015*; *Yadav et al., 2018*). The dramatic reduction in centromere size in *C. deuterogattii* may be attributable to loss of the RNAi pathway (*Farrer et al., 2015*; *Yadav et al., 2018*). The centromeres of *C. deuterogattii* consist of truncated transposable elements, and active transposable elements are missing throughout the genome (*Yadav et al., 2018*). This is in stark contrast to *C. neoformans,* which has active transposable elements in centromeric regions (*Dumesic et al., 2015*; *Janbon et al., 2014*; *Yadav et al., 2018*).

Neocentromeres are frequently formed near genomic repeats, yet *C. deuterogattii* lacks active transposons that might seed neocentromere formation. Thus, *C. deuterogattii* is a unique organism in which to study centromere structure and function. To elucidate centromeric organization, the native centromeres of chromosomes 10 and 9 were deleted, leading to characterization of the first neocentromeres in the *Basidiomycota* phylum of the fungal kingdom.

## Results

### Deletion of centromere 9 and 10 results in neocentromere formation

To determine if neocentromere formation occurs in the *C. deuterogattii* reference strain R265, either centromere 9 or 10 was deleted. Centromere 9 (*CEN9*) was deleted by CRISPR-Cas9-mediated transformation. Two guide RNAs flanking the centromere were used and *CEN9* was replaced with a *NAT* dominant drug-resistance gene by homologous recombination. Biolistic transformation was used to replace centromere 10 (*CEN10*) with either the *NAT* or *NEO* dominant drug-resistance gene via homologous recombination. Viable transformants with the correct integration and deletion were obtained and confirmed by 5' junction, 3' junction, loss of deleted regions, and spanning PCRs as well as Southern blot analysis for *cen10Δ* (*Figure 1—figure supplement 1*, *Figure 1—figure supplement 2*). Multiple independent *cen9Δ* and *cen10Δ* deletion mutants (*cen9Δ-A to -F* and *cen10Δ-A to -G*) were obtained from independent transformations. Pulsed-field gel electrophoresis (PFGE) confirmed that *cenΔ* mutants had a wild-type karyotype and that chromosome 9 and 10 remained linear, because a circular chromosome would not have entered the gel (*Figure 1—figure supplement 3*).

The formation of neocentromeres on chromosome 10 in *C. deuterogattii* was infrequent. A total of 99 independent biolistic transformations resulted in only seven confirmed *cen10Δ* mutants (7/21 total candidate transformants, 33% homologous integration), suggesting that *CEN10* deletion is lethal in most circumstances. In comparison, deletion of nonessential genes by homologous recombination in the *C. deuterogattii* R265 strain typically results in ~100 colonies with a high success rate (~80–90% homologous integration). We estimate that the likelihood of deleting a centromere and recovering a viable colony is at least 1000-fold lower than would be expected from the deletion of a non-essential gene. The deletion of *CEN9* was more efficient as this was mediated by CRISPR-Cas9 cleavage with two guide RNAs and a repair allele.

Chromatin immunoprecipitation of mCherry-CENP-A followed by high-throughput sequencing (ChIP-seq) for six *cen9Δ* (*-A to -F*) and seven *cen10Δ* mutants (*-A to -G*) was performed (*Figure 1*). Prior to the ChIP-seq experiment, all of the centromere deletion mutants were streak purified from single colonies. The sequence reads were mapped to a complete whole-genome assembly, followed by the normalization of the reads by subtraction of the input from the ChIPed sample (*Yadav et al., 2018*). To quantify the ChIP-seq data, the CENP-A-enriched regions were compared with the centromeres previously identified based on CENP-C enrichment. Both the CENP-A- and CENP-C-enriched peaks were congruent for all of the native centromeres (*Yadav et al., 2018*). This analysis identified 13 of the 14 native centromeres (*CEN1-8*, *CEN11-14* and depending on the centromere mutant either *CEN9* or *CEN10*), indicating that, as expected, the native centromere of chromosome 9 or 10 was missing in all of the *cen9Δ* and *cen10Δ* deletion mutants respectively (*Figure 1*). Instead, neocentromeres were observed.

Except for the neocentromere of isolate *cen10Δ-E*, the neocentromeres formed in close proximity to the native centromere (*CEN9* and *CEN10*). Almost all neocentromeres were shorter than the native centromere, with the exception of *cen10Δ-G* which was larger than native centromere 10 (*Table 1*).

In three of the independent *cen9Δ* mutants (*cen9Δ-B, -C* and *-E*), neocentromeres formed at the same chromosomal location (*Figure 1A*). Interestingly, two independent *cen10Δ* mutants (*cen10Δ-A* and *cen10Δ-C*) contained two CENP-A-enriched regions on chromosome 10, with a primary peak and a smaller secondary peak with reduced levels of CENP-A (1.3- to 1.75-fold lower) compared to the primary CENP-A peak (*Figure 1C*). The chromosomal location of the secondary peak was similar to the neocentromere of *cen10Δ-B* (which had only one neocentromere) (*Figure 1B*).

The two CENP-A-enriched regions suggest four possible models: 1) aneuploidy in which cells harbor two chromosomes, 2) a dicentric chromosome with two neocentromeres (neodicentric), 3) instability between two different neocentromere states (neocentromere switching), 4) or only one CENP-A-enriched region functions as a centromere and the second CENP-A-enriched region is not bound by the kinetochore (*Figure 1*).

The neocentromeres were located in unique, nonrepetitive sequences and were not flanked by repetitive regions. The GC content of neocentromeres is similar to the overall GC content of chromosome 9 and 10, whereas the native centromere has a lower GC content (*Table 1*). Comparing the reference genome with de novo genome assemblies of *cen10Δ-A*, *cen10Δ-B*, and *cen10Δ-E* confirmed that transposable elements did not enter these genomic regions during neocentromere

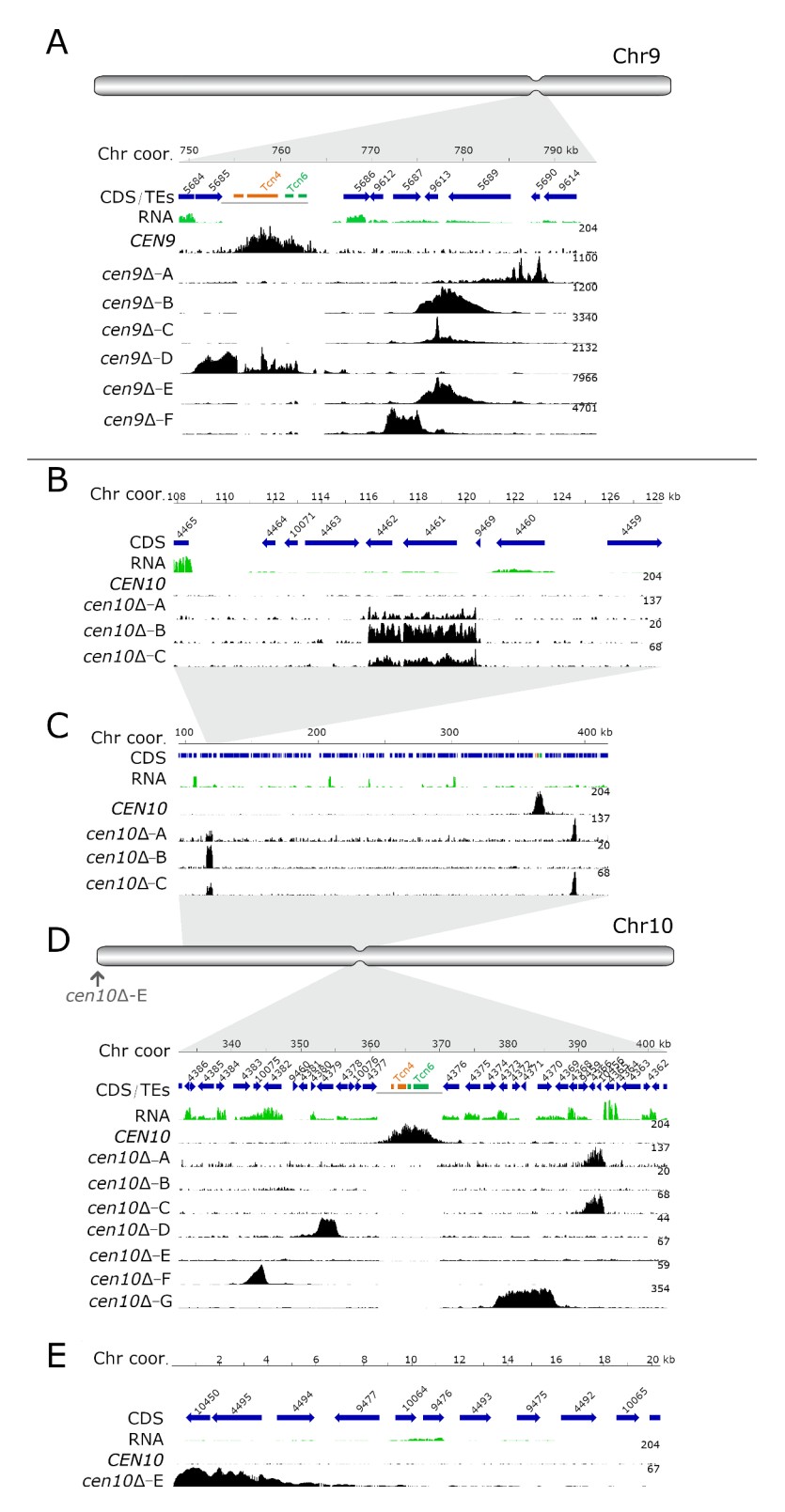

**Figure 1.** Centromere deletion leads to neocentromere formation. For each panel, the chromosome coordinates are indicated. Genes (CDS) are shown in blue arrows and the truncated transposable elements, located in the native centromere (*CEN9* or *CEN10*), are colored according to their class (Tcn4 in orange and Tcn6 in green). Previously generated RNA-sequencing obtained from wild-type cells was re-mapped and shown in green. In each panel, the wild-type CENP-A content is shown. In the wild type, CENP-A is only enriched at the native centromeres. For each *cen*Δ mutant, the

*Figure 1 continued on next page*

*Figure 1 continued*

neocentromeric region is shown by enrichment of CENP-A and the fold enrichment is indicated on the right of each ChIP-seq track. (**A**) Schematic full overview of chromosome 9, the indentation represents the native centromere 9 position. The light grey area points to the zoomed-in chromosomal region shown with the detailed view of the native centromere (*CEN9*) and the location of the *cen9Δ* mutant neocentromeres. Neocentromeres of *cen9Δ-B*, *cen9Δ-C* and *cen9Δ-E* formed at the same chromosomal location. The dark gray line, below the transposable elements, indicates the deleted region in the *cen9Δ* mutants (**B**) Detailed view of the neocentromere of *cen10Δ-B* and the secondary CENP-A peak of *cen10Δ-A* and *cen10Δ-C*. (**C**) Overview of the chromosomal 10 region spanning 100 to 410 kb. *cen10Δ-A* and *cen10Δ-C* have two regions enriched with CENP-A (primary and secondary). (**D**) Schematic full overview of the full chromosome 10, the indentation represents the chromosomal location of the native centromere (*CEN10*). The light grey areas point to the zoomed-in chromosomal regions shown in panel C and below. The neocentromere of *cen10Δ-E* is indicated with an arrow. Lower panel, detailed view of the native centromere (*CEN10*) and the neocentromeres formed in *cen10Δ-A*, *cen10Δ-C*, *cen10Δ-D*, *cen10Δ-F* and *cen10Δ-G* mutants. The dark gray line, below the transposable elements, indicates the deleted region in the *cen10Δ* mutants (**E**) Detailed view of the telocentric neocentromere of *cen10Δ-E*.

The online version of this article includes the following figure supplement(s) for figure 1:

**Figure supplement 1.** Confirmation of centromere 9 and 10 deletion by PCR.
**Figure supplement 2.** Centromere 10 is deleted in *cen10Δ* isolates.
**Figure supplement 3.** *cen9Δ* and *cen10Δ* mutants have a wild-type karyotype.
**Figure supplement 4.** ChIP-qPCR with additional kinetochore proteins.
**Figure supplement 5.** *cen10Δ* mutants have elongated cell morphology.

formation (*Supplementary file 1*). Instead of spanning repeats and transposable elements like the native centromeres, neocentromeres span genes.

All of the neocentromeres of chromosome 9 formed in a region within 26 kb of the chromosomal location of the native centromere 9 (*Figure 1A*). Interestingly, the neocentromeres of the independent *cen9Δ-B*, *cen9Δ-C*, and *cen9Δ-E* mutants all formed at the same chromosomal location and had the same length (4.41 kb); these neocentromeres spanned three genes. One gene was completely covered by CENP-A and this gene encodes a transglycosylase SLT domain-containing protein. The two other genes (a gene encoding a xylosylphosphotransferase and a gene encoding glutamate synthase (NADPH/NADH)) were partially covered with CENP-A. Mutant *cen9Δ-A* had a 3.87 kb long neocentromere located 26 kb 3′ to the native centromere and spanned two genes. The first gene was completely spanned by CENP-A and encodes an ESCRT-II complex subunit (Vps25) protein. The second gene was only partially covered and encodes an iron regulator protein. The neocentromere of *cen9Δ-D* was located directly to the left of the native centromere and was 4.37 kb in length. This neocentromere spanned two genes, coding for a hypothetical protein (92% covered by CENP-A) and a gene encoding for Derlin-2/3 that was completely covered by CENP-A. Lastly, the neocentromere of mutant *cen9Δ-F* was 3.83 kb in length and spanned one gene (encoding a xylosylphosphotransferase, *XPT1*), which was completely covered by CENP-A. This neocentromere was located 12 kb away (3′) from the native centromere 9.

Like the neocentromeres of *cen9Δ* mutants, the neocentromeres of *cen10Δ* mutants also spanned genes and interestingly, the kinetochore protein CENP-C was located inside the neocentromere of *cen10Δ-B* and in the secondary peak of *cen10Δ-A* and *-C* (*Table 1*, *Figure 1B*). The neocentromere in *cen10Δ-B* spanned 4.46 kb, was located 242 kb away from the 3′ region of the native *CEN10*, and was located 115 kb from the telomere (*Figure 1B*). In addition to the gene encoding CENP-C, the CENP-A-enriched region spanned a hypothetical protein (*Table 1*). The primary CENP-A-enriched region of *cen10Δ-A* and *cen10Δ-C* spanned a gene encoding a serine/threonine-protein phosphatase 2A activator 2 (*RRD2*) and a hypothetical protein (*Figure 1D*). This neocentromere spanned 2.85 kb and was located closer to the native *CEN10* (21 kb from the native centromere) than the neocentromere of *cen10Δ-B* and the secondary CENP-A peak of *cen10Δ-A* and *cen10Δ-C*. The neocentromere of *cen10Δ-D* was the smallest neocentromere (2.5 kb) and partially (88.4%) spanned a gene encoding a Ser/Thr protein kinase and formed 7.4 kb from the location of the native *CEN10* (*Figure 1D*). The neocentromere of *cen10Δ-E* spanned two hypothetical proteins, was 4.38 kb in length and was located directly adjacent to the right telomere (*Figure 1E*). Mutant *cen10Δ-F* had a neocentromere of 2.64 kb, which spanned one hypothetical gene completely and two genes (hypothetical and a hexokinase (*HXK1*)) partially; the neocentromere formed at a chromosomal location 20 kb 5′ of the native centromere (*Figure 1D*). The neocentromere of *cen10Δ-G* was the largest neocentromere with a CENP-A-enriched region of 7.97 kb, and was in fact larger than the native *CEN10*. This

**Table 1.** Genes located inside neocentromeres.

The chromosomal locations, sizes, and GC content (%) for the native centromere and *cen*Δ mutants are shown. For the neocentromeres, gene ID, predicted function, and the amount of CENP-A coverage are indicated.

| | Chr coor (bp) | Size (kb) | Size compared to native centromere (%) | GC % | Genes spanned by neocentromere | Gene ID | % covered by Neocentromere | Exons inside neocentromere |
|---|---|---|---|---|---|---|---|---|
| Native centromere 9 | Chr9:755,771–762,621 | 6.84 | - | 43.6 | - | - | - | - |
| cen9Δ-A | Chr9:785,352–789,247 | 3.87 | 56.6 | 46.1 | Escrt-II complex subunit (*VPS25*) | CNBG_5690 | 100 | |
| | | | | | Iron regulator 1 | CNBG_9614 | 14.6 | Last exon |
| cen9Δ-B | Chr9:775,164–780,756 | 4.41 | 64.5 | 46.6 | Xylosylphosphotransferase (*XPT1*) | CNBG_5687 | 6.9 | |
| | | | | | Transglycosylase SLT domain-containing protein | CNBG_9613 | 100 | |
| | | | | | Glutamate synthase (NADPH/NADH) | CNBG_5689 | 33.7 | |
| cen9Δ-C | Chr9:775,164–780,756 | 4.41 | 64.5 | 46.6 | Xylosylphosphotransferase (*XPT1*) | CNBG_5687 | 6.9 | |
| | | | | | Transglycosylase SLT domain-containing protein | CNBG_9613 | 100 | |
| | | | | | Glutamate synthase (NADPH/NADH) | CNBG_5689 | 33.7 | |
| cen9Δ-D | Chr9:750,902–755,294 | 4.37 | 63.9 | 41.9 | Hypothetical protein | CNBG_5684 | 92.8 | |
| | | | | | Derlin-2/3 | CNBG_5685 | 100 | |
| cen9Δ-E | Chr9:775,164–780,756 | 5.56 | 81.3 | 50 | Xylosylphosphotransferase (*XPT1*) | CNBG_5687 | 6.9 | |
| | | | | | Transglycosylase SLT domain-containing protein | CNBG_9613 | 100 | Last exon |
| | | | | | Glutamate synthase (NADPH/NADH) | CNBG_5689 | 33.7 | Last exon |
| cen9Δ-F | Chr9:771,614–775,469 | 3.83 | 56.0 | 51.5 | Xylosylphosphotransferase (*XPT1*) | CNBG_5687 | 100 | |
| Native centromere 10 | Chr10:362,876–369,657 | 6.77 | - | 42.6 | - | - | - | - |
| cen10Δ-A | Chr10:115,954–120,422 | 4.46 | 65.9 | 46.9 | *CENPC/MIF2* | CNBG_4461 | 88.3 | 1, 2, 3, 4 (only 5th is outside) |
| | | | | | Hypothetical protein | CNBG_4462 | 100 | |
| | Chr10:391,090–393,946 | 2.85 | 42.1 | 48.9 | Serine/threonine-protein phosphatase 2A activator 2(*RRD2*) | CNBG_9459 | 10.6 | Last exon (5th) |
| | | | | | Hypothetical protein | CNBG_4366 | 100 | |
| | | | | | Hypothetical protein | CNBG_4365 | 23.4 | Last exon (3th) |
| cen10Δ-B | Chr10:115,954–120,422 | 4.46 | 65.9 | 46.9 | *CENPC/MIF2* | CNBG_4461 | 88.3 | 1, 2, 3, 4 (only 5th is outside) |
| | | | | | Hypothetical protein | CNBG_4462 | 100 | |

*Table 1 continued on next page*

*Table 1 continued*

| | Chr coor (bp) | Size (kb) | Size compared to native centromere (%) | GC % | Genes spanned by neocentromere | Gene ID | % covered by Neocentromere | Exons inside neocentromere |
|---|---|---|---|---|---|---|---|---|
| *cen10Δ-C* | Chr10:115,954–120,422 | 4.46 | 65.9 | 46.9 | *CENPC/MIF2* | CNBG_4461 | 88.3 | 1, 2, 3, 4 (only 5th is outside) |
| | | | | | Hypothetical protein | CNBG_4462 | 100 | |
| | Chr10:391,090–393,946 | 2.85 | 42.1 | 48.9 | Serine/threonine-protein phosphatase 2A activator 2(*RRD2*) | CNBG_9459 | 10.6 | Last exon (5th) |
| | | | | | Hypothetical protein | CNBG_4366 | 100 | |
| | | | | | Hypothetical protein | CNBG_4365 | 23.4 | Last exon (3th) |
| *cen10Δ-D* | Chr10:352,648–355,154 | 2.51 | 37.1 | 48 | Ser/Thr protein kinase | CNBG_4379 | 88.4 | |
| *cen10Δ-E* | Chr10:1–4,385 | 4.38 | 64.7 | 53.2 | Hypothetical protein | CNBG_10450 | 100 | |
| | | | | | Hypothetical protein | CNBG_4495 | 100 | |
| *cen10Δ-F* | Chr10:342,517–345,159 | 2.64 | 39.0 | 45.5 | Hypothetical protein | CNBG_4383 | 18.6 | Last two exons |
| | | | | | Hypothetical protein | CNBG_10075 | 100 | |
| | | | | | Hexokinase (*HXK1*) | CNBG_4382 | 15.3 | Last three exons |
| *cen10Δ-G* | Chr10:378,389–386,366 | 7.97 | 117.7 | 46.5 | High osmolarity signaling protein (*SHO1*) | CNBG_4373 | 100 | |
| | | | | | Hypothetical protein | CNBG_4372 | 100 | |
| | | | | | Hypothetical protein | CNBG_4371 | 100 | |
| | | | | | Hypothetical protein | CNBG_4370 | 100 | |

neocentromere spanned four genes, including a gene coding for a high osmolarity protein (Sho1) and three genes coding for hypothetical proteins (*Figure 1D*).

To test if the kinetochore was binding to the CENP-A-enriched regions of chromosomes 9 and 10, and to validate if the neocentromeres were fully functional as centromeres, two additional kinetochore proteins were epitope-tagged with GFP. *cen9Δ* mutants were transformed with an overlap PCR product expressing *CENPC-GFP*. As the neocentromeres of three *cen10Δ* mutants spanned the gene encoding CENP-C, all *cen10Δ* mutants were transformed with an overlap PCR product, expressing Mis12-GFP. In addition to the *cen9Δ* and *cen10Δ* mutants, the wild type was transformed with constructs expressing Mis12-GFP and CENP-C-GFP, and these served as controls. ChIP-qPCRs for *cen9Δ* mutants, *cen10Δ* mutants, and wild-type strains with Mis12-GFP or CENP-C-GFP were performed (*Figure 1—figure supplement 4*). Because Mis12 is an outer kinetochore protein, the formaldehyde cross-linking was extended to 45 min (15 min was used for CENP-A and CENP-C) for this protein. For all qPCR analyses, the native centromere 6 *CEN6*) was used as an internal control and for each neocentromere specific primer pairs were designed. For *cen9Δ* and *cen10Δ* mutants a similar level of Mis12 or CENP-C enrichment at the neocentromeres and (*CEN6*) was observed. This suggested that the CENP-A-enriched regions of chromosome 9 of the *cen9Δ* mutants and chromosome 10 of *cen10Δ* mutants identified by ChIP-seq were functional centromeres and indeed neocentromeres (*Figure 1—figure supplement 4*).

Previously generated RNA sequence data were remapped to the *C. deuterogattii* reference strain R265 and analyzed to determine if the regions where neocentromeres formed in the *cenΔ* mutants were transcribed in the wild type (*Figure 1*; *Supplementary file 2*; *Schneider et al., 2012*). In the wild-type strain, all genes spanned by neocentromeres in the *cenΔ* mutants were expressed (*Figure 1*). However, the expression levels of the neocentromeric genes were lower than their neighboring genes. For example, the expression level of the gene (CNBG_5686) flanking native centromere 9 was three times higher than the genes spanned by neocentromeres in *cen9Δ* mutants (*Figure 1A*).

The same trend was observed in the *cen10Δ* mutants. Here, the expression level of the gene (CNBG_4365) 3' flanking the neocentromere (primary CENP-A peak) of *cen10Δ-A* and *cen10Δ-C* was more than six times higher than the genes spanned by the neocentromere (*Figure 1D*). Also, the neocentromere of *cen10Δ-D* is flanked by genes whose expression was two times higher than the genes spanned by the neocentromere (*Figure 1D*). This suggests that neocentromeres are formed in chromosomal regions with lower gene expression in *C. deuterogattii*. The majority of the genes (24/28) flanking native centromeres are transcribed in the direction towards the native centromere. All of the neocentromeres observed span one or more genes and most of the flanking genes are transcribed in the direction away from the neocentromere.

The expression levels of the neocentromeric genes in *cen9Δ* and *cen10Δ* mutants were assayed by qPCR (*Figure 2*). The neocentromeric genes of chromosome 9 were normalized to actin. To compensate for the ploidy levels of chromosome 10 in *cen10Δ* mutants, a housekeeping gene located on chromosome 10 was used to normalize the expression of genes spanned by neocentromeres located on chromosome 10. The expression levels of the CENP-A-associated neocentromeric genes were all found to be similar to the wild-type strain (*Figure 2*).

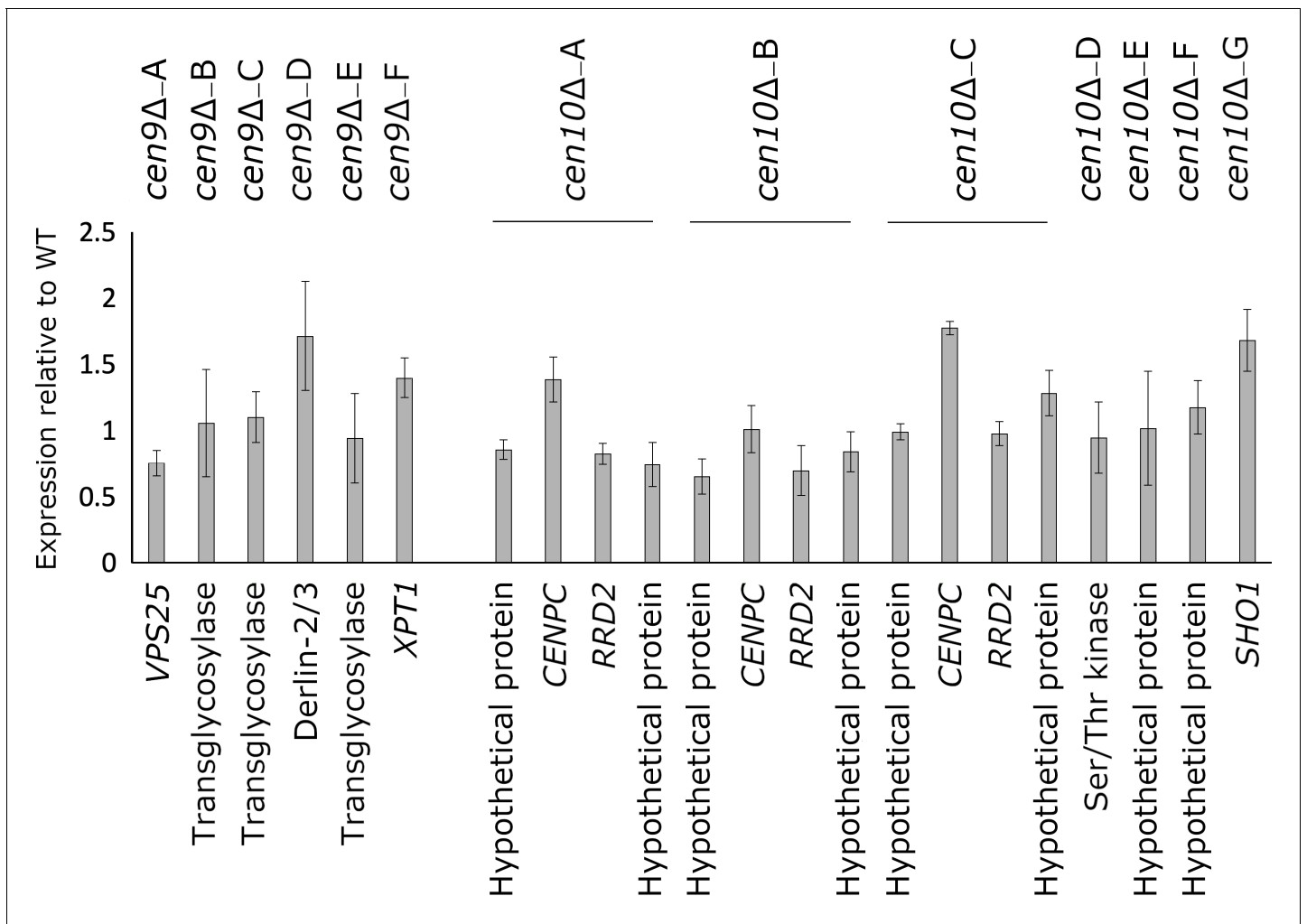

**Figure 2.** Expression of neocentromeric genes. Expression of the neocentromeric genes was assessed by qPCR for all *cenΔ* mutants and expression is shown as Log2$^{ΔΔCt}$. For *cen10Δ-A, cen10Δ-B* and *cen10Δ-C,* two genes were selected from each neocentromeric region, all other *cenΔ* mutants are represented by one gene spanned by CENP-A. *cen10Δ-B* has only one CENP-A-enriched region, and in this case, the genes located within primary peak of *cen10Δ-A* and *cen10Δ-C* served as controls. The qPCRs of *cen10Δ* mutants are normalized with a housekeeping gene located on chromosome 10. The qPCRs of *cen9Δ* mutants are normalized with actin. Error bars show standard deviation.

## Neocentromere formation can reduce fitness

*cen10Δ* mutants were noted to grow more slowly than wild type. To investigate this, the growth of *cen10Δ* and wild-type strains was measured during the course of a 22-hour cell growth experiment (*Figure 3A*). The majority of *cen10Δ* mutants exhibited slower growth rates compared to the wild-type parental strain R265. Six of seven *cen10Δ* mutants exhibited significant fitness defects compared to the wild-type strain, with doubling times ranging from 101 to 111 min compared to 81 min for the wild type (*Figure 3B*). In contrast, one mutant, which has a telocentric neocentromere (*cen10Δ-E*), grew similarly to the wild type and had a similar doubling time (84 min for the mutant vs 81 min for the wild-type strain). Compared to the wild type, *cen10Δ* mutants with increased doubling times produced smaller colonies during growth on non-selective media (*Figure 1—figure supplement 5C*).

To compare fitness, a competition assay was performed with 1:1 mixtures of wild-type and *cen9Δ* or *cen10Δ* mutants grown in liquid YPD medium (*Figure 3C*). With no growth defect, the expectation was that the wild-type strain and centromere deletion mutants would grow at the same growth rate, resulting in a 1:1 ratio. In fact, fewer *cen10Δ* cells were found in the population after growth in competition with the wild-type strain, and this observation is consistent with the slower doubling time of *cen10Δ* mutants resulting in reduced fitness compared to wild type (*Figure 3*). Compared to

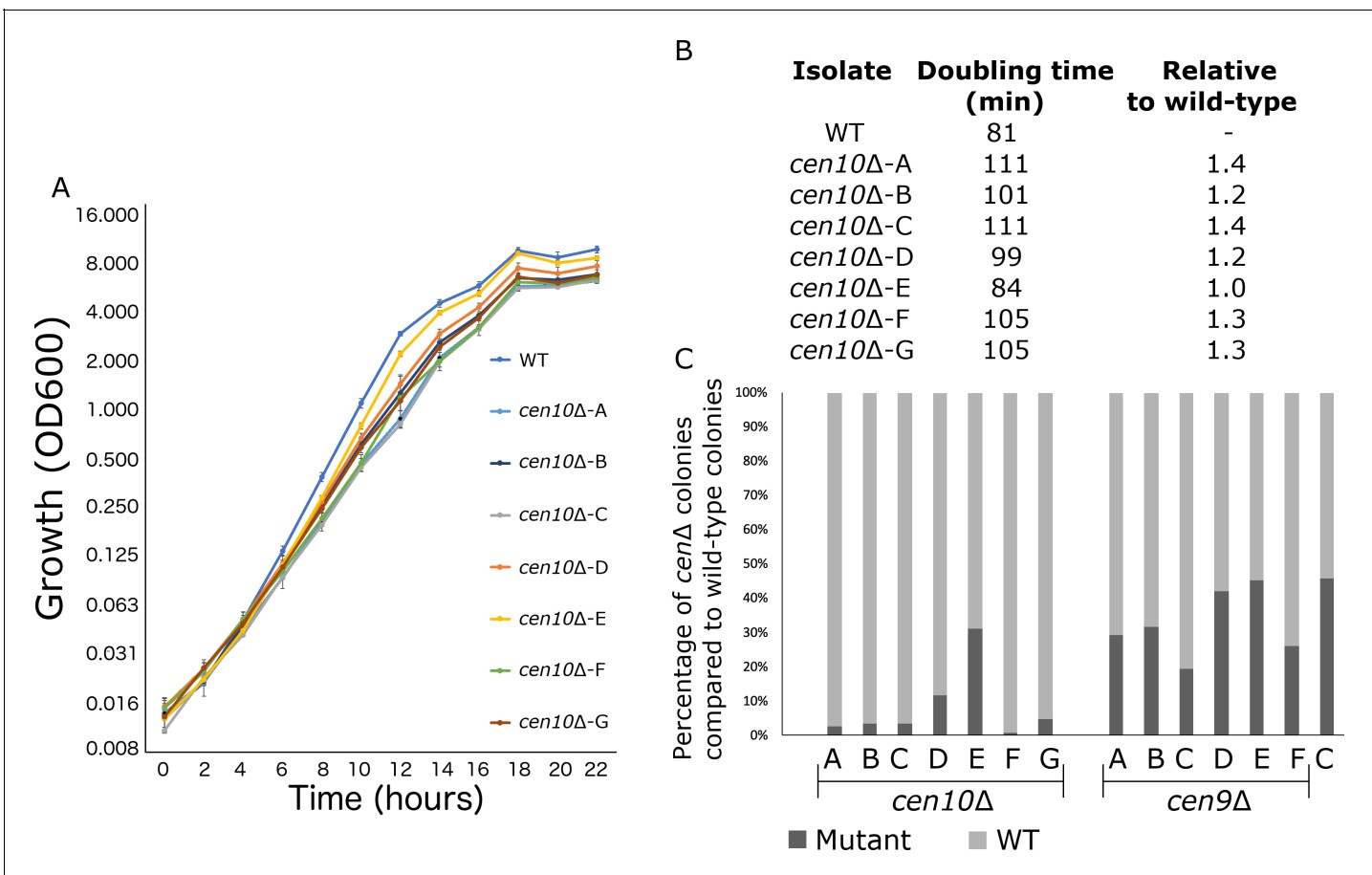

**Figure 3.** *cen10Δ* mutant strains have reduced fitness compared to the wild-type strain. (A) Six out of seven *cen10Δ* mutants had a longer doubling time and slower growth than the wild-type strain. In contrast *cen10Δ-E* grows similarly to the wild type. Error bars show standard deviation. (B) Doubling times and fold change compared to wild type are shown. (C) Competition assays with the wild type and *cen9Δ* and *cen10Δ* mutant strains. Mixed cultures (1:1) were grown overnight and plated with and without selection agents. After four days, colonies were counted and the percentage of *cenΔ* mutants (black) and wild type (grey) in each culture was plotted. As a control (C) a wild-type strain with a *NAT* marker was mixed with the wild type. The online version of this article includes the following figure supplement(s) for figure 3:

**Figure supplement 1.** *cen10Δ* mutants with chromosomal fusion have a wild-type growth rate.

the wild-type cells, there were fewer *cen9Δ* mutant cells in the population. However, the number was closer to a 1:1 ratio (*Figure 3C*). The ratio of the *cen9Δ* mutants in the population was similar to the ratio of the *cen10Δ-E* mutant, which had a wild-type growth rate. Due to this observation, we hypothesize that the growth rate of the *cen9Δ* mutants is similar to wild type.

## *cen10Δ* isolates are aneuploid

Because deletion of a centromere could lead to defects in chromosome segregation, *cenΔ* mutants were assessed for aneuploidy (*Figure 4*). Overall, *cen10Δ* mutants exhibited a mixture of large and small colony sizes during growth on YPD medium at 37°C, while *cen9Δ* mutants exhibited a uniform, wild-type like, colony size (*Figure 1—figure supplement 5*).

Aneuploidy in *C. neoformans* often leads to a similar mixed colony size phenotype as that observed in the *cen10Δ* mutants (*Sun et al., 2014*). To exacerbate the aneuploidy-associated slow growth phenotype, four *cen10Δ* mutants were grown at elevated temperature (37°C), causing these isolates to produce smaller, growth-impaired and larger, growth-improved colonies (*Figure 3—figure supplement 1*). Three small and two large colonies were selected from each isolate and whole-genome analysis was performed based on Illumina sequencing. Sequences were mapped to the reference R265 genome, revealing that the small colonies were indeed aneuploid (*Figure 4A*). The small colonies of *cen10Δ-B* and *cen10Δ-C* had ploidy levels for chromosome 10 in the range of 1.25- to 1.36-fold higher compared to the other 13 chromosomes, which suggested that only a proportion of the cells (25% to 36%), were aneuploid (*Figure 4B*). The remainder of the genome was euploid. Chromosome 10 of the small colonies derived from isolate *cen10Δ-A* and *cen10Δ-E* exhibited ploidy levels ranging from 1.1- to 1.14-fold, reflecting less aneuploidy. Importantly, for all of the large colonies derived from isolates *cen10Δ-A*, *cen10Δ-B*, *cen10Δ-C*, and *cen10Δ-E* the fold coverage of chromosome 10 was restored to the wild-type euploid level (1.0 fold compared to wild type). The ploidy levels of chromosome 10 were 1-fold for all of the large colonies compared to wild type, indicating that the ploidy level of chromosome 10 of the large colonies was restored to euploid.

## *cen10Δ* chromosome is rescued by chromosome fusion

Based on whole-genome sequencing and PFGE analysis, fusion of *cen10Δ* chromosome 10 to other chromosomes was a common event in the large colonies (*Figure 5*, *Figure 6*). Whole-genome sequence analysis revealed that sequences corresponding to the 3' subtelomeric region of chromosome 10 (including one gene) were absent in the sequences obtained from all of the large colonies analyzed (*Figure 4—figure supplement 1A*). In addition, the large colonies of *cen10Δ-A* were missing sequences for two genes in the 5' subtelomeric region of chromosome 4 (*Figure 4—figure supplement 1B*). Large colonies of *cen10Δ-B* were missing 18.5 kb at the 5' subtelomere of chromosome 7 (including eight genes) (*Figure 4—figure supplement 1C*). The large colonies of *cen10Δ-E* lacked a small part of one gene in the 3' subtelomeric region of chromosome 1. In total, of the 14 subtelomeric genes that were lost in these three chromosome-fusion isolates, ten encoded hypothetical proteins and four encoded proteins with predicted functions. BlastN analysis in the de novo genome assemblies of the large colonies confirmed that the subtelomeric regions were not located on minichromosomes or inserted in other chromosomes. Seven genes have homologs in *C. neoformans* and are present in *C. neoformans* deletion libraries (*Liu et al., 2008*; *Supplementary file 3*). This observation suggested that either subtelomeric deletions occurred, or that chromosomal fusions led to the loss of subtelomeric regions. Notably, sequences from the small colonies spanned the entire genome with no evidence of these subtelomeric deletions (*Figure 4—figure supplement 1F*).

We hypothesized that the subtelomeric gene loss was due to chromosomal fusion and tested this hypothesis with de novo genome assemblies and PFGE (*Figure 5* and *Figure 6*). Based on de novo genome assemblies for the large colonies of *cen10Δ-A*, *cen10Δ-B*, and *cen10Δ-E*, chromosome 10 fused with chromosome 4, 7, or 1, respectively (*Figure 5* and *Supplementary file 3*). In the large colony of *cen10Δ-A* (*cen10Δ-A-L1*), the fusion occurred between chromosome 10 and chromosome 4 (*Figure 5A*). Chromosomal fusion led to the loss of the CNBG_10211 gene (on chromosome 4), and the fusion junction was within the CNBG_6174 gene of chromosome 4. For *cen10Δ-B-L1*, the chromosomal fusion occurred between chromosomes 10 and 7 (*Figure 5B*). Seven genes of chromosome 7 were lost in the fused chromosome. The chromosome fusion junction was intergenic on

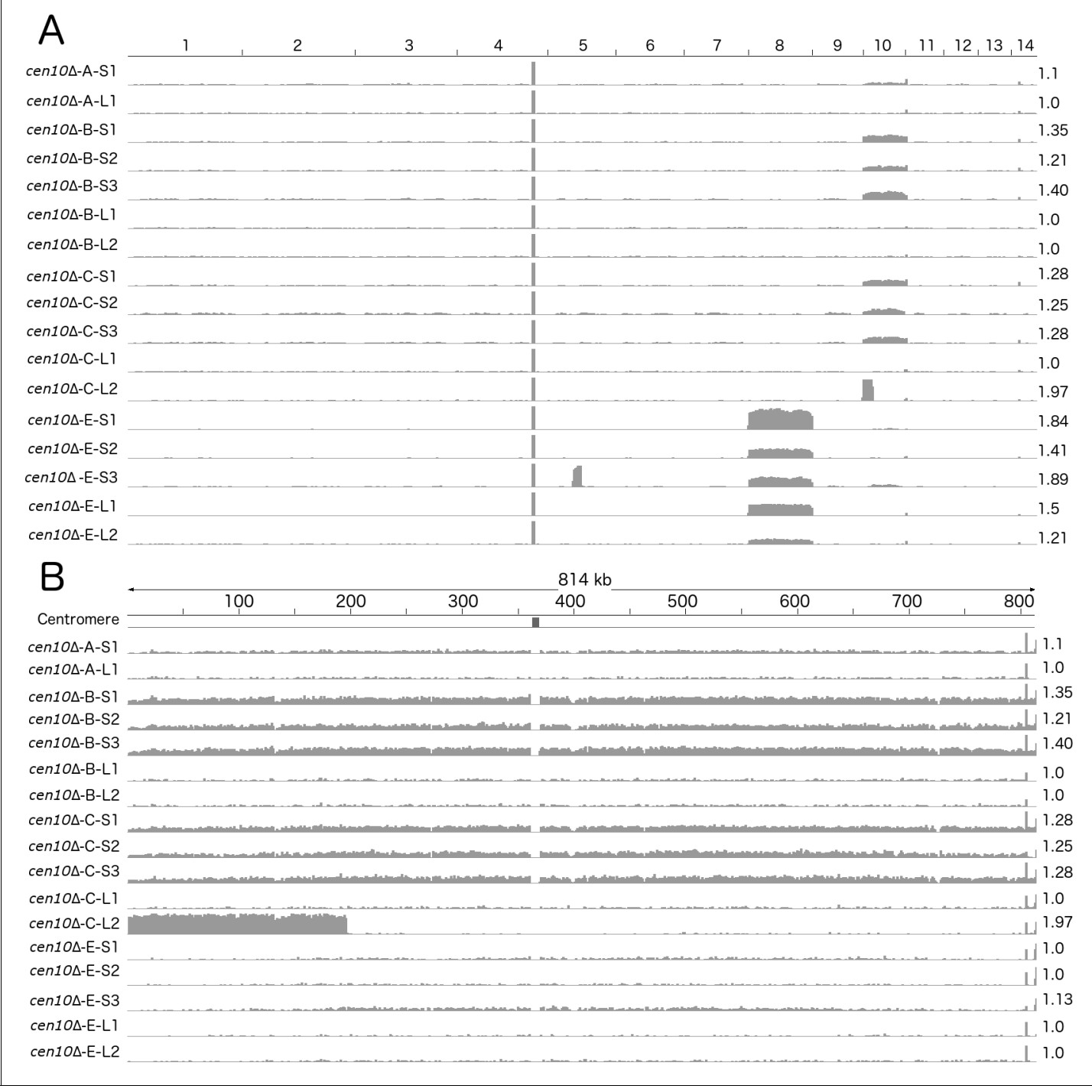

**Figure 4.** *cen10Δ* mutants are aneuploid. The whole genomes of small and large colonies derived from four *cen10Δ* mutants were sequenced and read coverage (corresponding to ploidy levels) was plotted. Small colonies of *cen10Δ* mutants were partially aneuploid for chromosome 10, while the large colonies are euploid. (**A**) Genome-wide read depth coverage for small and large colonies. On the right, the fold coverage for the highest ploidy level is indicated for each sample. For example, chromosome 10 of *cen10Δ-B-S1* had an aneuploidy level of 1.35-fold compared to the wild-type strain. Chromosome 4 had a small region with increased read depth due to the ribosomal rDNA gene cluster and was excluded from the analysis. Chromosome 8 of *cen10Δ-E* was duplicated. In addition, *cen10Δ-E-S3* had an additional duplicated region of 162 kb of chromosome 5 that spans the sequence of native centromere 5. (**B**) Detailed view of read depth of chromosome 10. As in panel A, read depth is indicated on the right. The native centromeric location is shown by a black square. Due to the deletion of centromere 10, the location of the native centromere lacks sequence reads for each sample.

The online version of this article includes the following figure supplement(s) for figure 4:

*Figure 4 continued on next page*

*Figure 4 continued*

**Figure supplement 1.** Deletion within subtelomeric regions in chromosome fusion isolates.

chromosome 7. *cen10Δ-E-L1* was due to a chromosomal fusion between chromosomes 10 and 1 (*Figure 5C*). The fusion was intragenic for both chromosomes. The fusion point occurred in CNBG_6141 on chromosome 10 and CNBG_10308 on chromosome 1.

Because all of the large *cen10Δ* colonies had chromosome 10 fusions, we examined the fusion location on chromosome 10 in detail. The fusions occurred 1.7, 0.3, and 3.6 kb from the chromosome 10 gene CNBG_6142, respectively (*Figure 5—figure supplement 1*). The fusion occurred in unique DNA sequences and was not flanked by repetitive regions. The overlapping region between chromosome 10 and the fused chromosome was at most 6 bp, suggesting that these fusions

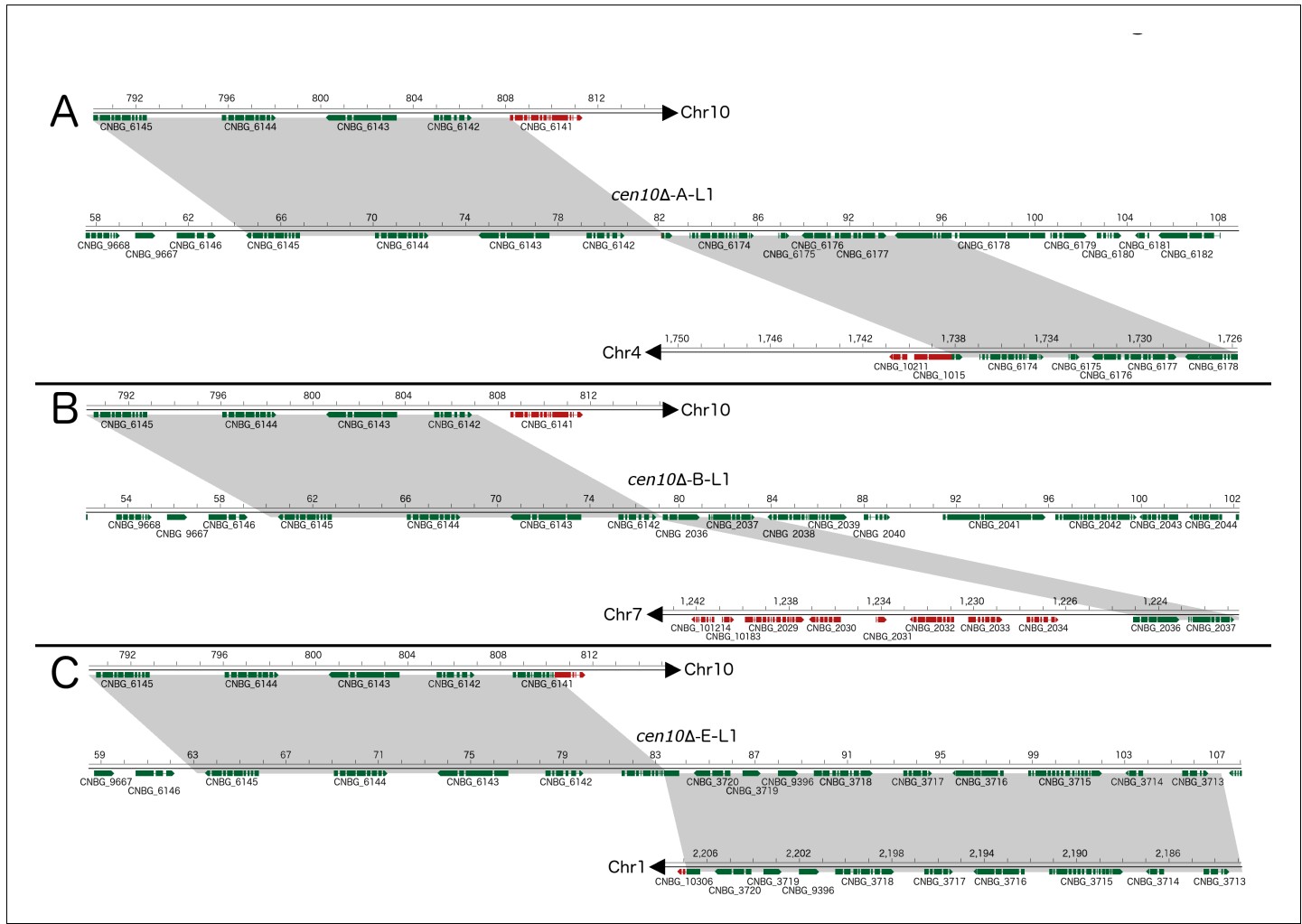

**Figure 5.** *cen10Δ* mutants undergo chromosome fusion leading to improved fitness at 37°C. Chromosomal fusions were studied in detail for three *cen10Δ* mutants restored to wild-type growth levels at 37°C (large colonies). After chromosome fusion, the fused chromosomes of *cen10Δ-A-L* and *cen10Δ-B-L* lost the gene CNBG_6141, which is located in the 3' subtelomeric region of chromosome 10. Genes present in the fused chromosome are depicted in green, and genes lost after chromosome fusion are indicated in red. Gray highlights indicate regions present in both the parental and fused chromosomes. Each fusion occurred in a unique nonrepetitive region. (**A**) *cen10Δ-A-L1*, the fusion occurred between chromosome 10 and chromosome 4. (**B**) In *cen10Δ-B-L1*, chromosomal fusion occurred between chromosomes 10 and 7. (**C**) *cen10Δ-E-L1* chromosomal fusion occurred between chromosomes 10 and 1.

The online version of this article includes the following figure supplement(s) for figure 5:

**Figure supplement 1.** Chromosome fusion in large *cen10Δ* colonies.

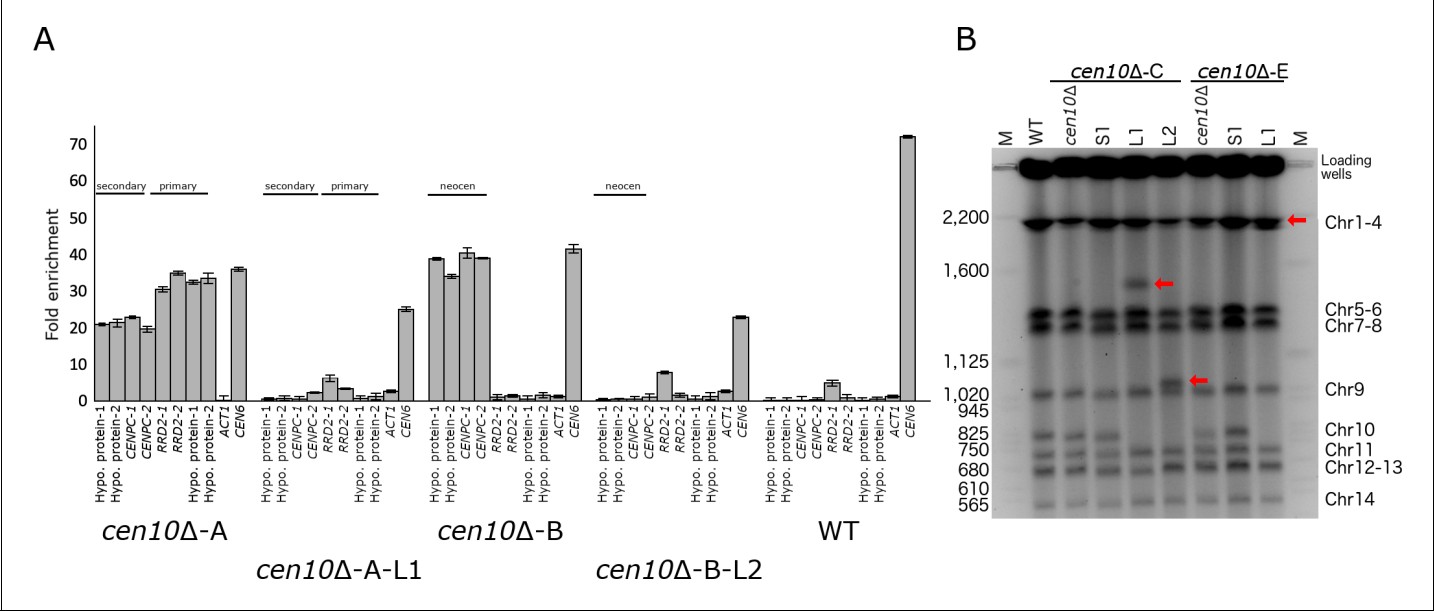

**Figure 6.** Chromosome fusion results in neocentromere inactivation and karyotype reduction. (A) Neocentromeres are inactive after chromosomal fusion. For each neocentromere two qPCR primer pairs located in genes spanned by the neocentromere in *cen10Δ-A* and *cen10Δ-B* mutant were used in a ChIP-qPCR experiment. Analyzed is the CENP-A enrichment of 1) a *cen10Δ* mutant, 2) a large colony derived from the *cen10Δ* mutant, and 3) the wild-type strain. Centromere 6 (*CEN6*) was included as a positive control, and actin was included as a negative control. Data are shown for *cen10Δ-A*, *cen10Δ-A-L1*, *cen10Δ-B*, *cen10Δ-B-L2*, and wild type. For *cen10Δ-A* and *cen10Δ-A-L1* mutants, the chromosomal regions investigated are indicated according to the primary and secondary CENP-A peaks of the *cen10Δ-A* mutant. The *cen10Δ-B* mutant has only one CENP-A-enriched region which co-localized with the secondary CENP-A peak of *cen10Δ-A* and this region is labeled with neocen in *cen10Δ-B* and *cen10Δ-B-L1*. Error bars show standard deviation. (B) PFGE analysis shows that the band corresponding to chromosome 10 was lost in the large colonies and instead larger bands appear due to the fusion of chromosome 10 with other chromosomes. *cen10Δ* deletion mutants and small colonies derived from 37°C show a wild-type karyotype. Chromosome 10 of the large colonies was fused to chromosome 13, 10, or 1, respectively. Due to limitations of PFGE conditions, the chromosome 10–chromosome 1 fusion did not separate from chromosomes 2, 3, and 4. The positions of the fused chromosomes are indicated with arrows.

occurred via microhomology-mediated end joining (MMEJ) (also known as alternative nonhomologous end-joining [Alt-NHEJ]).

Chromosome fusion may result in loss of the neocentromeres, and the kinetochore may bind to the native centromere of the fused chromosome and function as the active centromere. This hypothesis was tested by performing ChIP-qPCR for CENP-A binding (*Figure 6A*). For each neocentromere (either of the *cen10Δ-A* or *cen10Δ-B* mutant), CENP-A enrichment was tested with four primer sets located in the neocentromere. The CENP-A enrichment for these four locations was tested in 1) the initial *cen10Δ* mutant, 2) a large colony derived from a specific *cen10Δ* mutant and 3) wild type (*Figure 6A*). As expected, the ChIP-qPCR analysis showed CENP-A enrichment for the neocentromeres of the initial *cen10Δ-A* and *cen10Δ-B* mutants. The neocentromeric regions of *cen10Δ-A* and *cen10Δ-B* were not enriched with CENP-A in the wild-type strain, showing there was no occupancy by CENP-A prior to neocentromere formation. For all analyzed *cen10Δ* chromosome 10 fusion isolates, the neocentromeres were not CENP-A-associated, and were similar to the wild-type background levels. Therefore, the neocentromeres were no longer active in the chromosome fusion strains (*Figure 6A*). This suggests that the native centromere of the fusion partner of chromosome 10 (i.e., chromosome 1, 4, or 7) was the active centromere of the Chr10-Chr1, Chr10-Chr4, and Chr10-Chr7 fusions.

In addition to *cen10Δ-A*, *-B*, and *-E* mutants, whole-genome sequencing was performed for two large colonies of *cen10Δ-C*. Although it was not possible to identify the chromosome fusion based on whole-genome sequencing data for either of the large colonies of the *cen10Δ-C* mutant, PFGE analysis showed that *cen10Δ-C-L1* had a fusion between chromosomes 10 and 13 (*Figure 6b*). *cen10Δ-C-L2* had read coverage of 1.99-fold for a region of ~200 kb of chromosome 10 (*Figure 4B*). The rest of chromosome 10 was euploid, suggesting that the ~200 kb region was duplicated and was either a single chromosome or fused to another chromosome in this isolate. PFGE analysis

suggested that this fragment was duplicated on chromosome 10, resulting in a larger chromosome (*Figure 6B*). In contrast to the other fused chromosomes, this chromosomal fragment did not fuse to a chromosome with a native centromere, and the fact that the mutant still exhibited a fitness defect was consistent with this interpretation. The larger chromosome was euploid, suggesting that the unstable neocentromere(s), rather than causing aneuploidy, resulted in a fitness cost in this isolate.

## Discussion

### Composition of neocentromeres in *C. deuterogattii*

The native centromeres of *C. deuterogattii* are found in repetitive regions and are flanked by, but do not contain, protein-encoding genes (*Yadav et al., 2018*). By contrast, neocentromeres of *C. deuterogattii* span genes, lack repetitive elements, and like the native centromeres, are flanked by genes. In general (with one exception), the neocentromeres of *C. deuterogattii* are significantly shorter than the native centromeres, whereas most neocentromeres in other species have similar lengths as the native centromeres.

Native centromeres of *S. pombe* have a central core that is enriched with CENP-A and flanked by repetitive pericentric regions (*Ishii et al., 2008*). While neocentromere formation in *S. pombe* favors repeats in the pericentric regions, neocentromere formation is possible without the repetitive pericentric regions (*Ishii et al., 2008*). The majority of the neocentromeres in *C. albicans* and chickens are formed close to native centromeres due to seeding of CENP-A that is located near the native centromere (the so-called CENP-A cloud) (*Ketel et al., 2009*; *Shang et al., 2013*). Our results and the earlier reports discussed, suggest that the chromosomal location of the native centromere is the main determinant of neocentromere formation. One exception was the neocentromere of *cen10Δ-E*, which directly flanked the left telomere. Interestingly, this was the only *cen10Δ* mutant that had a growth rate similar to wild type.

Several *C. deuterogattii* neocentromeres formed in the same location; however, there is no apparent consensus between the different regions occupied by different neocentromeres. A similar trend has been observed in neocentromere formation in *C. albicans* (*Ketel et al., 2009*). Evolutionary new centromeres (ECNs) in the largest crucifer tribe Arabideae originated several times independently and are located in the same chromosomal location (*Mandáková et al., 2020*). Our results suggest that neocentromeres form by mechanisms that do not rely on nearby transposable elements/repeats to initiate de novo centromere assembly.

### Neocentromeric genes are expressed

Neocentromeres induced in several species can span genes, resulting in silencing or reduced gene expression. For example, all genes within five independent neocentromeres in *C. albicans* that spanned nine genes were suppressed (*Burrack et al., 2016*). In *S. pombe*, neocentromeres span genes that are only expressed in response to nitrogen starvation in the wild-type strain, and neocentromere formation silences these genes during nitrogen starvation (*Ishii et al., 2008*). The native centromere 8 of rice contains an approximately 750 kb CENP-A-enriched region with four genes that are expressed in both leaf and root tissues of three closely related species (*Fan et al., 2011*; *Nagaki et al., 2004*). Neocentromeres of rice span genes that are expressed at similar levels as in the wild type (*Zhang et al., 2013*). Chicken neocentromeres have been induced on chromosome Z or 5 (*Shang et al., 2013*). Chromosome Z neocentromeres span eight genes, but in wild-type cells only *MAMDC2* is expressed during normal growth. The other seven genes were either not expressed at any detectable level in all tested developmental stages or were only expressed during early embryonic stages (*Shang et al., 2013*). When a neocentromere formed, expression of the MAMDC2-encoding gene was reduced 20- to 100-fold. Chromosome 5 of chickens is diploid, and neocentromeres on this chromosome span genes that are expressed. The hypothesis behind this phenomenon is that one allele functions as a centromere, while the other allele codes for the genes.

Because the *cen10Δ* mutants of *C. deuterogattii* were aneuploid, the expression of genes spanned by chromosome 10 neocentromeres was normalized to expression levels of a housekeeping gene located on chromosome 10. The expression of genes enriched for CENP-A chromatin is similar to that of wild type, and if the allelic hypothesis, like in chickens, were valid, the expectation would be a 60% reduction in expression levels. The genes spanned by neocentromeres of *cen9Δ* mutants

are also expressed at wild-type levels. As the *cen9Δ* mutants have uniform, wild-type colony sizes, the ploidy levels of these mutants were not tested and we hypothesize that these mutants are haploid/euploid.

Genes contained in regions in which *C. deuterogattii* neocentromeres formed in *cenΔ* mutants were actively expressed in the wild-type strain, and this is similar to human neocentromeres that can form in regions with or without gene expression (*Alonso et al., 2010*; *Marshall et al., 2008*). However, we have identified chromosomal regions that lack gene expression on chromosomes 9 and 10, although these regions were not close to the native centromere.

Of the *C. deuterogattii* genes spanned by the neocentromere region, one encodes the kinetochore component CENP-C. Several independent biolistic transformations were performed to delete the gene encoding CENP-C, but all attempts were unsuccessful. This suggests that *CENPC* is an essential gene. In fission yeast, deletion of the gene encoding the CENP-C homolog *Cnp3* was lethal at 36°C, but mutants were still viable at 30°C (*Suma et al., 2018*). However, CENP-A was mislocalized in the *cnp3Δ* mutants. Another gene partially located inside a *C. deuterogattii* neocentromere encodes the serine/threonine-protein phosphatase 2A activator 2 (*RRD2*). The *RRD2* homolog is not essential in *S. cerevisiae* (*Higgs and Peterson, 2005*).

Compared with other haploid fungi, the neocentromeric genes of *C. deuterogattii* are similar to the native centromeric genes of the haploid plant pathogenic fungus *Zymoseptoria tritici*. *Z. tritici* has short regional centromeres with an average size of 10.3 kb, and 18 out of 21 native centromeres have a total of 39 expressed genes (*Schotanus et al., 2015*).

### *cen10Δ* mutants with two CENP-A-enriched regions

The appearance of two CENP-A-enriched regions of *C. deuterogattii cen10Δ* mutants could be explained in a few ways. First, neocentromere formation could lead to a dicentric chromosome 10 in which the centromeres may differ in functional capacity. Dicentric chromosomes are not by definition unstable, for example the dominant-negative mutation of the mammalian telomere protein TRF2 results in chromosome fusions, leading to the formation of dicentric chromosomes (*Stimpson et al., 2010*). The formation of dicentric chromosomes occurred in 97% of the fused mammalian chromosomes, which were stable for at least 180 cell divisions (*Stimpson et al., 2010*). Several microscopic studies showed that chromosomes with two regions of centromere-protein enrichment are stable (*Higgins et al., 2005*; *Stimpson et al., 2012*; *Stimpson et al., 2010*; *Sullivan and Willard, 1998*). This suggests that a dineocentric chromosome 10 could be stable in the population. Second, the two CENP-A-enriched peaks could be the result of a mixed population and either due to an unstable primary neocentromere and/or aneuploidy. The primary neocentromere could be associated with the majority of the cells, whereas the secondary CENP-A peak would be only found in a small number of cells (and the primary neocentromere is lost in these isolates). This is reflected by lower CENP-A enrichment for the secondary peak, and the hypothesis of putative dicentrics is due to a mixture of alleles in the population. Third, the neocentromeres could be unstable, which could lead to the formation of two CENP-A-enriched regions with centromere function switching between the regions. However, our data would argue against this latter model. Prior to the ChIP-seq analysis of the *cen10Δ* mutants, colonies were isolated by streak purification (eight times), suggesting that the presence of two distinct CENP-A peaks occurs continuously.

### *cen10Δ* mutants are partially aneuploid

Neocentromere formation in chickens results in a low number of aneuploid cells (*Shang et al., 2013*). Based on whole-genome sequencing of a population of cells, the *C. deuterogattii cen10Δ* isolates are partially aneuploid for chromosome 10. For fully aneuploid isolates, the coverage of Illumina reads is expected to be 2-fold; the *cen10Δ* isolates with two CENP-A peaks showed aneuploidy levels up to 1.28-fold or were even euploid. This suggests that, like the chicken neocentromeric isolates, only a small number of cells in a population of *C. deuterogattii cen10Δ* isolates are aneuploid.

### *cen10Δ* mutants have reduced fitness

In *C. albicans*, deletion of centromere 5 results in neocentromere formation, and these isolates have fitness similar to the wild-type strain (*Ketel et al., 2009*). Similar results were reported for

neocentromeres in chicken and *S. pombe*, in which strains with neocentromeres or chromosome fusion have a growth rate similar to the wild-type strain (*Ishii et al., 2008*; *Shang et al., 2013*).

If centromere deletions occurred in nature, we hypothesize that the wild type would outcompete all of the *cen∆* isolates. The virulence of the *cen∆* mutants was not assayed. Based on reduced fitness of the *cen∆* mutants, we hypothesize that pathogenicity of the *cen∆* mutants would be lower than the wild type. However, when chromosome fusion occurs the growth rate is restored to a near wild-type level and we hypothesize that the isolates with 13 chromosomes could have virulence similar to the wild type. Several genes were lost due to the fusion events in the *cen∆* mutants; to our knowledge these lost genes have not been associated with pathogenicity of *C. deuterogattii*.

## Neocentromere stains exhibit impaired growth and chromosome fusion restores wild-type growth at elevated temperatures

Deletion of a centromere in *S. pombe* leads to either neocentromere formation or chromosome fusion due to a noncanonical homologous recombination pathway (*Ishii et al., 2008*; *Ohno et al., 2016*). This is in contrast to neocentromere formation in *C. deuterogattii*, which results in 100% neocentromere formation. Based on PFGE analysis, the karyotype of the *cen∆* isolates is wild type at 30°C, but chromosome fusion can occur at 37°C within the *cen10∆* mutants and lead to improved growth at 30°C.

The location of the *cen10∆* neocentromere had no apparent influence on the ability to undergo chromosome fusion as we have shown with a telocentric neocentromere, a dicentric mutant, and a neocentromere located 118 kb away from the telomere.

The fused chromosomes have no or only short homology at the breakpoints that is insufficient for homologous recombination, suggesting that the chromosome fusions arise via MMEJ. Future experiments to test this hypothesis could involve deleting genes involved in the MMEJ pathway, such as *CDC9* and *DNL4* (*Sinha et al., 2016*).

A prominent chromosome fusion occurred during the speciation of humans. Compared to other great apes, humans have a reduced karyotype, which is due to the fusion of two ancestral chromosomes that resulted in chromosome 2 in modern humans, Denisovans, and Neanderthals (*Miga, 2017*). Human chromosome2 still harbors signatures of telomeric repeats at the fusion point (interstitial telomeric sequences [ITS]), suggesting that this chromosome is derived from a telomere-telomere fusion. By synteny analysis, the inactive centromere of chimpanzee chromosome 2b can be identified on human chromosome 2, and there are relics of α satellite DNA at this now extinct centromere (*Miga, 2017*). Moreover, a dominant-negative mutation of the human telomeric protein TRF2 leads to telomere-telomere fusions, mainly between acrocentric chromosomes (*Stimpson et al., 2010*; *van Steensel et al., 1998*). In the fungal species *Malassezia*, chromosome breakage followed by chromosome fusion has led to speciation (*Sankaranarayanan et al., 2020*). The short regional centromeres (3–5 kb) are fragile and this led most likely to chromosome reduction. By contrast in *C. deuterogattii*, the chromosomes involved in chromosomal fusion of the *cen10∆* mutants were all metacentric, and fusion occurred in non-telomeric sequences.

Another example of telomeric fusions is the presence of ITS regions in several genomes. In budding yeast, the experimental introduction of an ITS into an intron of the *URA3* gene resulted in four classes of chromosome rearrangements, including: 1) inversion, 2) gene conversion, 3) mini-chromosome formation due to deletion or duplication, and 4) mini-chromosome formation due to translocation (*Aksenova et al., 2013*). Based on our de novo genome assemblies of the *C. deuterogattii* large-colony *cen10∆* mutants, chromosome fusions occurred with no signs of chromosome rearrangements. Thus, these chromosome fusions did not produce ITS regions, which would otherwise destabilize the genome.

## Conclusions

Our work shows that, like in other model systems, neocentromeres can be induced in *C. deuterogattii*. However, *C. deuterogattii* neocentromeres have several unique characteristics, such as spanning genes whose expression is unaffected by centromere assembly. In some instances, deletion of *CEN10* led to chromosome fusion, resulting in enhanced fitness and leading to inactivation of the neocentromere. Presumably, deletion of other centromeres could be carried out, leading to a *C.*

*deuterogattii* strain with only one or a few chromosomes, as was recently reported in *S. cerevisiae* (*Luo et al., 2018*; *Shao et al., 2018*).

# Materials and methods

Key Resources Table Template and Guidelines.

## Key resources table

| Reagent type (species) or resource | Designation | Source or reference | Identifiers | Additional information |
|---|---|---|---|---|
| Genetic reagent *Cryptococcus deuterogattii* | R265 | This study | | R265 expressing *mCherry-CENPA* |
| Genetic reagent *Cryptococcus deuterogattii* | cen10△-A | This study | | R265 centromere 10 deletion mutant with expressing *mCherry-CENPA* |
| Genetic reagent *Cryptococcus deuterogattii* | cen10△-B | This study | | R265 centromere 10 deletion mutant with expressing *mCherry-CENPA* |
| Genetic reagent *Cryptococcus deuterogattii* | cen10△-C | This study | | R265 centromere 10 deletion mutant with expressing *mCherry-CENPA* |
| Genetic reagent *Cryptococcus deuterogattii* | cen10△-D | This study | | R265 centromere 10 deletion mutant with expressing *mCherry-CENPA* |
| Genetic reagent *Cryptococcus deuterogattii* | cen10△-E | This study | | R265 centromere 10 deletion mutant with expressing *mCherry-CENPA* |
| Genetic reagent *Cryptococcus deuterogattii* | cen10△-F | This study | | R265 centromere 10 deletion mutant with expressing *mCherry-CENPA* |
| Genetic reagent *Cryptococcus deuterogattii* | cen10△-G | This study | | R265 centromere 10 deletion mutant with expressing *mCherry-CENPA* |
| Genetic reagent *Cryptococcus deuterogattii* | cen10△-A-S1 | This study | | Small colony derived from R265 centromere 10A deletion mutant with expressing *mCherry-CENPA* |
| Genetic reagent *Cryptococcus deuterogattii* | cen10△-A-L1 | This study | | Large colony derived from R265 centromere 10A deletion mutant with expressing *mCherry-CENPA* |
| Genetic reagent *Cryptococcus deuterogattii* | cen10△-B-S1 | This study | | Small colony derived from R265 centromere 10B deletion mutant with expressing *mCherry-CENPA* |
| Genetic reagent *Cryptococcus deuterogattii* | cen10△-B-S2 | This paper | | Small colony derived from R265 centromere 10B deletion mutant with expressing *mCherry-CENPA* |
| Genetic reagent *Cryptococcus deuterogattii* | cen10△-B-S3 | This study | | Small colony derived from R265 centromere 10B deletion mutant with expressing *mCherry-CENPA* |
| Genetic reagent *Cryptococcus deuterogattii* | cen10△-B-L1 | This study | | Large colony derived from R265 centromere 10B deletion mutant with expressing *mCherry-CENPA* |

*Continued on next page*

*Continued*

| Reagent type (species) or resource | Designation | Source or reference | Identifiers | Additional information |
|---|---|---|---|---|
| Genetic reagent *Cryptococcus deuterogattii* | cen10△-B-L2 | This study | | Large colony derived from R265 centromere 10B deletion mutant with expressing *mCherry-CENPA* |
| Genetic reagent *Cryptococcus deuterogattii* | cen10△-C-S1 | This study | | Small colony derived from R265 centromere 10C deletion mutant with expressing *mCherry-CENPA* |
| Genetic reagent *Cryptococcus deuterogattii* | cen10△-C-S2 | This study | | Small colony derived from R265 centromere 10C deletion mutant with expressing *mCherry-CENPA* |
| Genetic reagent *Cryptococcus deuterogattii* | cen10△-C-S3 | This study | | Small colony derived from R265 centromere 10C deletion mutant with expressing *mCherry-CENPA* |
| Genetic reagent *Cryptococcus deuterogattii* | cen10△-C-L1 | This study | | Large colony derived from R265 centromere 10C deletion mutant with expressing *mCherry-CENPA* |
| Genetic reagent *Cryptococcus deuterogattii* | cen10△-C-L2 | This study | | Large colony derived from R265 centromere 10C deletion mutant with expressing *mCherry-CENPA* |
| Genetic reagent *Cryptococcus deuterogattii* | cen10△-E-S1 | This study | | Small colony derived from R265 centromere 10E deletion mutant with expressing *mCherry-CENPA* |
| Genetic reagent *Cryptococcus deuterogattii* | cen10△-E-S2 | This study | | Small colony derived from R265 centromere 10E deletion mutant with expressing *mCherry-CENPA* |
| Genetic reagent *Cryptococcus deuterogattii* | cen10△-E-S3 | This study | | Small colony derived from R265 centromere 10E deletion mutant with expressing *mCherry-CENPA* |
| Genetic reagent *Cryptococcus deuterogattii* | cen10△-E-L1 | This study | | Large colony derived from R265 centromere 10E deletion mutant with expressing *mCherry-CENPA* |
| Genetic reagent *Cryptococcus deuterogattii* | cen10△-E-L2 | This study | | Large colony derived from R265 centromere 10E deletion mutant with expressing *mCherry-CENPA* |
| Genetic reagent *Cryptococcus deuterogattii* | cen9△-A | This study | | R265 centromere 9 deletion mutant with expressing *mCherry-CENPA* |
| Genetic reagent *Cryptococcus deuterogattii* | cen9△-B | This study | | R265 centromere 9 deletion mutant with expressing *mCherry-CENPA* |
| Genetic reagent *Cryptococcus deuterogattii* | cen9△-C | This study | | R265 centromere 9 deletion mutant with expressing *mCherry-CENPA* |
| Genetic reagent *Cryptococcus deuterogattii* | cen9△-D | This study | | R265 centromere 9 deletion mutant with expressing *mCherry-CENPA* |
| Genetic reagent *Cryptococcus deuterogattii* | cen9△-E | This study | | R265 centromere 9 deletion mutant with expressing *mCherry-CENPA* |

*Continued on next page*

*Continued*

| Reagent type (species) or resource | Designation | Source or reference | Identifiers | Additional information |
|---|---|---|---|---|
| Genetic reagent *Cryptococcus deuterogattii* | *cen9△-F* | This study | | R265 centromere 9 deletion mutant with expressing *mCherry-CENPA* |
| Genetic reagent *Cryptococcus deuterogattii* | R265 *MIS12* | This study | | R265 expressing *GFP-MIS12* and *mCherry-CENPA* |
| Genetic reagent *Cryptococcus deuterogattii* | *cen10△-A MIS12* | This study | | R265 Centromere 10 mutant with expressing *GFP-MIS12* and *mCherry-CENPA* |
| Genetic reagent *Cryptococcus deuterogattii* | *cen10△-B MIS12* | This study | | R265 Centromere 10 mutant with expressing *GFP-MIS12* and *mCherry-CENPA* |
| Genetic reagent *Cryptococcus deuterogattii* | *cen10△-C MIS12* | This study | | R265 Centromere 10 mutant with expressing *GFP-MIS12* and *mCherry-CENPA* |
| Genetic reagent *Cryptococcus deuterogattii* | *cen10△-D MIS12* | This study | | R265 Centromere 10 mutant with expressing *GFP-MIS12* and *mCherry-CENPA* |
| Genetic reagent *Cryptococcus deuterogattii* | *cen10△-E MIS12* | This study | | R265 Centromere 10 mutant with expressing *GFP-MIS12* and *mCherry-CENPA* |
| Genetic reagent *Cryptococcus deuterogattii* | R265 *CENPC* | This study | | R265 with expressing *GFP-CENPC* and *mCherry-CENPA* |
| Genetic reagent *Cryptococcus deuterogattii* | *cen9△-A CENPC* | This study | | R265 Centromere 9 mutant with expressing *GFP-CENPC* and *mCherry-CENPA* |
| Genetic reagent *Cryptococcus deuterogattii* | *cen9△-B CENPC* | This study | | R265 Centromere 9 mutant with expressing *GFP-CENPC* and *mCherry-CENPA* |
| Genetic reagent *Cryptococcus deuterogattii* | *cen9△-C CENPC* | This study | | R265 Centromere 9 mutant with expressing *GFP-CENPC* and *mCherry-CENPA* |
| Genetic reagent *Cryptococcus deuterogattii* | *cen9△-D CENPC* | This study | | R265 Centromere 9 mutant with expressing *GFP-CENPC* and *mCherry-CENPA* |
| Genetic reagent *Cryptococcus deuterogattii* | *cen9△-E CENPC* | This study | | R265 Centromere 9 mutant with expressing *GFP-CENPC* and *mCherry-CENPA* |
| Antibody | Anti-mCherry antibody (Rabbit polyclonal) | Abcam | Cat. no. ab183628 | ChIP (1/5000) |
| Antibody | Anti-GFP (Rabbit polyclonal) antibody | Abcam | Cat. no. ab290 | ChIP (5 µg for 1 µg of chromatin) |
| Other | Dynabeads Protein A for Immunoprecipitation | Invitrogen | Cat. no. 10001D | ChIP (20 µl per 500 µl fraction) |
| Software, algorithm | Bowtie2 | *Langmead, 2010* | | |
| Software, algorithm | Spades | *Bankevich et al., 2012* | | |
| Software, algorithm | IGV | *Thorvaldsdóttir et al., 2013* | | |

*Continued on next page*

*Continued*

| Reagent type (species) or resource | Designation | Source or reference | Identifiers | Additional information |
|---|---|---|---|---|
| Software, algorithm | HISAT2 | *Pertea et al., 2016* | | |
| Sequence-based reagent | List of primers used in this study | Sigma | | In *Supplementary file 4* |

## Strains, primers, and culture conditions

Primers are listed in *Supplementary file 4*. Strains used in this study are listed in *Supplementary file 5*. All strains were stored in glycerol at −80°C, inoculated on solid YPD (yeast extract, peptone, and dextrose) media, and grown for two days at 30°C. Liquid YPD cultures were inoculated from single colonies of solid media and grown, while shaking, at 30°C overnight.

## Genetic manipulations

DNA sequences (1 to 1.5 kb) of the *CEN10*-flanking regions were PCR-amplified with Phusion High-Fidelity DNA Polymerase (NEB, Ipswich MA, USA). Flanking regions were fused on both sides of either the *NEO* or *NAT* dominant selectable marker via overlap PCR, conferring G418 or nourseothricin resistance, respectively. Deletion of *C. deuterogattii CEN10* was achieved through homologous recombination via biolistic introduction of an overlap-PCR product as previously described (*Billmyre et al., 2017*; *Davidson et al., 2002*). Deletion of *CEN9* was performed by CRISPR-CAS9 mediated transformation with two guide RNAs flanking *CEN9* and homologous recombination was mediated by the introduction of an overlap PCR product as previously described (*Fan and Lin, 2018*). Transformants were selected on YPD medium containing G418 (200 µg/mL) or nourseothricin (100 µg/mL).

Subsequently, the 5' junction, 3' junction, and spanning PCR and Southern blot analyses were performed to confirm the correct replacement of *CEN10* by the appropriate drug resistance marker. To identify centromeres, the gene CNBG_0491, which encodes CENP-A, was N-terminally fused to the gene encoding the fluorescent mCherry protein by overlap PCR, and *C. deuterogattii* strains were biolistically transformed as previously described (*Billmyre et al., 2017*). A subset of *cen9Δ* mutants were biolistically transformed with an overlap PCR product containing *CENPC* C-terminally fused with *GFP*. As three *cen10Δ* mutants have a neocentromere that spans the gene encoding CENPC, a subset of *cen10Δ* mutants were transformed instead with an overlap PCR product containing *MIS12* C-terminally fused with *GFP*. Both PCR products encoding *CENPC-GFP* and *MIS12-GFP* were randomly integrated in the genome and confirmed by a PCR spanning either *CENPC-GFP* or *MIS12-GFP*.

## Growth and competition assays

Three replicate cultures for seven independent *cen10Δ* deletion mutants and the wild-type strain were grown in liquid YPD at 30°C overnight. Cells were diluted to an $OD_{600}$ of 0.01 and grown in 50 mL YPD at 30°C. The $OD_{600}$ of the triplicate cultures was measured every two hours with a Smart-Spec 3000 (BioRad) until stationary phase was reached (T = 22 hr).

For competition assays, three independent replicate cultures (*cen9Δ*, *cen10Δ*, control, and wild type) were grown overnight in 8 mL YPD. Subsequently, the cell density of the cultures was determined using a hemocytometer. For each independent *cenΔ* deletion mutant, 500,000 cells were co-cultured in a 1:1 ratio with wild-type cells. After 24 hr, the cultures were inoculated on 1) a YPD plate to determine the total colony-forming units (CFUs) and 2) a YPD plate containing G418 or nourseothricin to calculate the proportion of *cen10Δ* mutant CFUs compared to the wild-type CFUs. Plates were incubated at 30°C and the colonies were counted after 4 days. The cell morphology of >1000 cells of the wild type and of five *cen10Δ* mutant strains was analyzed, and the number of elongated cells was quantified (*Figure 1—figure supplement 5*).

## Whole-genome sequencing, read mapping for aneuploidy/RNA-seq, and de novo genome assemblies

Genomic DNA was isolated using the CTAB protocol and sent to the Duke University Sequencing and Genomic Technologies Shared Resource facility for library preparation and Illumina sequencing. Sequencing was performed with a HiSeq 4000 sequencer, and 150 bp paired-end reads were generated. The resulting DNA sequence reads were trimmed, quality-filtered, and subsequently mapped with Bowtie2 to a complete PacBio, Nanopore-based, Illumina Pilon-corrected, whole-genome assembly of the *C. deuterogattii* R265 reference genome (version R265_fin_nuclear). Reads were visualized with IGV (*Langmead, 2010*; *Quinlan and Hall, 2010*; *Thorvaldsdóttir et al., 2013*; *Yadav et al., 2018*). Previously generated RNA sequencing reads (NCBI, SRA: SRR5209627) were remapped to the *C. deuterogattii* R265 reference genome by HISAT2 according to the default settings (*Schneider et al., 2012*; *Pertea et al., 2016*).

Genomes were de novo assembled with Spades using the default conditions (*Bankevich et al., 2012*). Genome assemblies were confirmed with PCRs using primers flanking the chromosome fusions and the PCR products obtained span the chromosomal fusions (*Figure 5—figure supplement 1*). The read coverage at chromosome fusions was analyzed and compared to the average read coverage of the contig (*Figure 5—figure supplement 1*).

## Chromatin immunoprecipitation (ChIP) followed by high-throughput sequencing or qPCR

ChIP analyses were performed as previously described with minor modifications (*Schotanus et al., 2015*; *Soyer et al., 2015*). In short, 500 mL YPD cultures (1000 ml YPD for Mis12 ChIPs) were grown overnight at 30°C, after which 37% formaldehyde was added to a final concentration of 0.5% for crosslinking. The cultures were then incubated for 15 min, formaldehyde was quenched with 2.5 M glycine (1/20 vol), and cells were washed with cold PBS. The crosslinking time of Mis12-GFP tagged isolates was extended to 45 min. Cells were resuspended in chromatin buffer (50 mM HEPES-NaOH, pH 7.5; 20 mM NaCl; 1 mM Na-EDTA, pH 8.0; 1% [v/v] Triton X-100; 0.1% [w/v] sodium deoxycholate [DOC]) containing protease inhibitors (cOmplet Tablets, mini EDTA-free EASYpack, Roche), followed by homogenization by bead beating with a miniBead beater (BioSpec products) using 18 cycles of 1.5 min on and 1.5 min off. The supernatant containing chromatin was sheared by sonication (24 cycles of 15 s on, 15 s off, burst at high level) (Bioruptor UCD-200, Diagenode). Chromatin was isolated by centrifugation, and the supernatant was divided into a sample fraction and a sonication control. The sample fraction was precleared with protein-A beads (1 to 3 hr) and subsequently divided into two aliquots. One tube served as the input control, and a mCherry or GFP antibody (MCherry: ab183628, Abcam, GFP: ab290, Abcam) was added to the remaining half of the sample. The samples were incubated overnight at 4°C and then processed according to a previously published protocol (*Soyer et al., 2015*). After completing the ChIP experiment, the samples were analyzed by ChIP-qPCR or sent to the Duke University Sequencing and Genomic Technologies Shared Resource facility for library preparation and Illumina sequencing. Samples *cen10Δ-A*, *cen10Δ-B* and *cen10Δ-C* were sequenced with a HiSeq 2500 sequencer, and single reads of 50 bp were obtained. All other ChIP-seq samples were sequenced with a NovaSeq 600 sequencer and 50 bp PE reads were obtained. For each centromere mutant and the wild type, a ChIPed and input sample were sequenced. Reads were mapped to the reference genome, similar to the whole-genome sequencing reads. To analyze the ChIP-seq data the ChIPed sample was normalized with the input sample and visualized with the IGV viewer.

## RNA isolation and qPCRs

Cells were grown in an overnight culture of 25 mL YPD at 30°C. RNA was isolated with TRIzol LS (Thermo Fisher Scientific) according to the manufacturer's instructions. Subsequently, cDNA was synthesized with the SuperScriptFirst-Strand Synthesis System (Thermo Fisher Scientific) according to the manufacturer's instructions. qPCRs were performed in triplicate with Brilliant III Ultra-Fast SYBR Green qPCR Master Mix (Agilent Technologies) on an ABI 1900HT qPCR machine.

## Pulsed-field gel electrophoresis (PFGE)

Isolation of whole chromosomes and conditions for PFGE and chromoblot analysis were performed as previously described (*Findley et al., 2012*).

## Acknowledgements

We thank Tom Petes, Beth Sullivan, Kaustuv Sanyal, Sue Jinks-Robertson, Vikas Yadav, Shelby Priest, Shelly Clancey, and Inge van der Kloet for comments on the manuscript. We would like to thank all of the members of the Heitman and Sanyal labs who contribute to the bi-weekly Skype meeting. These studies were supported by NIH/NIAID grants R01 AI050113-15 and R37 MERIT award AI039115-22 to JH. JH is co-director and fellow of the CIFAR program Fungal Kingdom: Threats and Opportunities.

## Additional information

### Funding

| Funder | Grant reference number | Author |
| --- | --- | --- |
| NIH | AI050113-15 | Joseph Heitman |
| NIH | AI039115-22 | Joseph Heitman |
| CIFAR | Fungal Kingdom: Threats and Opportunities | Joseph Heitman |

The funders had no role in study design, data collection and interpretation, or the decision to submit the work for publication.

### Author contributions

Klaas Schotanus, Conceptualization, Data curation, Software, Formal analysis, Validation, Investigation, Visualization, Methodology, Writing - original draft, Project administration, Writing - review and editing; Joseph Heitman, Conceptualization, Resources, Supervision, Funding acquisition, Project administration, Writing - review and editing

### Author ORCIDs

Klaas Schotanus (iD) https://orcid.org/0000-0002-0974-2882
Joseph Heitman (iD) https://orcid.org/0000-0001-6369-5995

### Decision letter and Author response

Decision letter https://doi.org/10.7554/eLife.56026.sa1
Author response https://doi.org/10.7554/eLife.56026.sa2

## Additional files

### Supplementary files

• Supplementary file 1. Neocentromeres are not enriched with transposable elements. To exclude the possibility that transposable elements were deposited into the neocentromeres, BlastN searches in a database with de novo genome assemblies of *cen10Δ-A*, *cen10Δ-B*, *cen10Δ-C*, and *cen10Δ-E* were performed. As input the homologous wild-type sequence of the chromosomal location of the neocentromeres was used. All neocentromeres of the tested *cen10Δ* mutants have the same length as the homologous sequence in the wild -type.

• Supplementary file 2. Neocentromeric regions are expressed in the wild-type strain. Expression levels of genes where neocentromeres formed in the *cenΔ* mutants were analyzed in the R265 wild-type strain. Previously generated RNA sequencing data were remapped to the R265 reference genome, and the expression levels (FPKM) were analyzed for the native genes in each region wherein a neocentromere was formed in the *cenΔ* mutants. Expression analysis of several housekeeping genes was included for control purposes, and the median RNA expression level of all genes

located on chromosomes 9 and 10 are listed. Expression levels of genes located in the subtelomeric regions were also analyzed.

• Supplementary file 3. Genes located in subtelomeric regions of chromosomes 1, 4, 7, and 10. We have indicated whether each gene located in the subtelomeric regions was lost in the chromosomal fusion of the large-colony *cen10Δ* mutants. For each gene in the subtelomeric region of chromosome 1, 4, 7, and 10, several characteristics, such as the chromosomal location, putative function, and the presence of putative *C. neoformans* homologs in existing mutant libraries, are indicated. There is no correlation between the loss of genes with or without predicted function.

• Supplementary file 4. Primers used in this study. For each primer, the lab identifier, purpose, and sequence are shown.

• Supplementary file 5. Strains used in this study. For each strain used in this study, the lab strain identifier, description, and parental strain are indicated.

• Transparent reporting form

## Data availability

ChIP and whole-genome sequencing reads and de novo genome assemblies were deposited under NCBI BioProject Accession ID: PRJNA511460.

The following dataset was generated:

| Author(s) | Year | Dataset title | Dataset URL | Database and Identifier |
|---|---|---|---|---|
| Schotanus K | 2020 | Neocentromere formation in *Cryptococcus deuterogattii* | http://www.ncbi.nlm.nih.gov/bioproject/?term=PRJNA511460 | NCBI BioProject, PRJNA511460 |

The following previously published dataset was used:

| Author(s) | Year | Dataset title | Dataset URL | Database and Identifier |
|---|---|---|---|---|
| Universidade Federal do Rio Grande do Sul | 2017 | *Cryptococcus gattii* R265 mRNA RNA-Seq | https://www.ncbi.nlm.nih.gov/sra/SRR5209627 | NCBI Sequence Read Archive, SRR5209627 |

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
