## [Decision Letter]

**Acceptance summary:**

In this manuscript the authors analyze the consequences of centromere deletions on chromosomes 9 and 10 in the fungal pathogen *Cryptococcus deuterogattii*. The authors identify the formation on neocentromeres at two different locations on chromosome 10. Elevated growth temperature selects for chromosome fusions of chromosome 10 with another chromosome involving inactivation of the neocentromere. The chromosomes 9 with neocentromeres do not display this behavior. Notably, the neocentromeric genes are constitutively expressed. The events described in this work present a mechanism that could explain the changes in genome organization within the Cryptococcus pathogenic species complex.

**Decision letter after peer review:**

[Editors’ note: the authors submitted for reconsideration following the decision after peer review. What follows is the decision letter after the first round of review.]

Thank you for submitting your work entitled "Centromere deletion in *Cryptococcus deuterogattii* leads to neocentromere formation and chromosome fusions" for consideration by *eLife*. Your article has been reviewed by three peer reviewers, including Wolf-Dietrich Heyer as the Reviewing Editor and Reviewer #1, and the evaluation has been overseen by a Senior Editor.

Our decision has been reached after consultation between the reviewers. Based on these discussions and the individual reviews below, we regret to inform you that your work will not be considered further for publication in *eLife*.

As you can see from the reviews, there was a span of opinions, which converged at the end of our discussion. We all agreed that the manuscript is interesting and that the process of neocentromere formation is an important topic. We appreciate the amount of work required to adequately analyze and describe these events, but the analysis offers little new insights into the mechanism of neocentromere formation beyond the description. As *eLife* wants to limit the required revisions to take no more than 2 months, we could not see how the major points listed in the reviews could be addressed in that time span. For this reason, we decided on a rejection. However, if you elect to address the comments in a future submission to *eLife*, I would be happy to serve as a reviewing editor again.

Reviewer #1:

This is a well written manuscript analyzing the consequences of centromere 10 deletion in the fungal pathogen *Cryptococcus deuterogattii*. The authors identify the formation on neocentromeres at two different locations on chromosome 10. Elevated growth temperature selects for chromosome fusions of chromosome 10 with another chromosome involving inactivation of the neocentromere. The events described in this work present a mechanism that could explain the changes in genome organization within the Cryptococcus pathogenic species complex.

The evidence presented supports the conclusions, and my main comment is that the analysis of the 2 isolates with 2 mapped neocentromeres remains inconclusive.

Essential revision:

1) The authors identify 2 isolates that contain 2 neocentromeres on chromosome 10 as defined by CENP-A ChIP. They present 4 models to explain the observation of having 2 CENP-A enrichment loci in a single chromosome: 1) aneuploidy, 2) a dicentric, 3) instability and neo-centromere switching, 4) only one CENPA locus being a functional centromere. The authors show that only a minority fraction of cells is aneuploid based on sequence read analysis. In the Discussion, they argue against a switching model, and discuss the possibility of the other two model. This is not very satisfying, and a ChIP experiment using a kinetochore component could be conducted to assess the functionality of both CENPA loci to assemble a kinetochore. There may be alternative routes to resolve this.

Reviewer #2:

*Cryptococcus deuterogattii* has regional centromeres and lacks active transposons. In this paper, authors deleted centromere 10 and examined chromosomes in surviving clones after centromere deletion (*Cen10Δ* mutants). This deletion caused neocentromere formation in two particular sites. Unlike neocentromere observed in other organisms, neocentromere formation in *C. deuterogattii* did not prevent expression of gene that cover neocentromeres. *Cen10Δ* mutants showed growth defects and aneuploidy. If growth temperature was increased, neocentromeres were inactivated and chromosome-fusion occurred. Finally authors speculated that these observations might occur under natural situation during speciation.

This is potentially interesting study, but work in this study is relatively descriptive. My major concern is unclearness how authors' observation in this study will be generalized. Authors performed various nice experiments and data are clear, however, I am not sure whether chromosomes with a neocentromere always get fusion with other chromosomes with growth fitness. In addition, it is also unclear whether cells with a neocentromere always show aneuploid. I agree that neocentromeres in this study are relatively weak compared with native centromeres, but I am not sure whether neocentromeres in *C. deuterogattii* are always weak or not. If authors show an evidence that observations in this study frequently occur or they can clearly explain about generality of this work, my mind will be changed.

Reviewer #3:

Centromeres vary in size and sequence between organisms. In most organisms studied, the centromere is marked by the presence of the histone H3 variant, CENPA. Accumulated evidence indicates that CENPA is the epigenetic mark that specifies centromere identity. Consistent with an epigenetic mode of centromere propagation in which sequence does not dominate, neocentromeres can arise on chromosomal regions which have no homology to centromere sequences. Many naturally-occurring neocentromeres have been reported and neocentromeres have been experimentally induced in several organisms, including fission yeast, Candida and chicken.

This manuscript describes the isolation and characterization of neocentromere strains in the human pathogen *Cryptococcus deuterogattii*, using an approach similar to previous studies in other species including *Candida albicans*. The authors delete the centromere of chromosome 10 and recover rare survivors which they go on to show by CENPA ChIP-seq have acquired either one or two neocentromeres on chromosome 10. They report that this has little impact on underlying gene expression. By whole genome sequencing the authors provide evidence that the strains have a degree of aneuploidy, and are growth impaired. By incubation of neocentromere strains at 37^o^C, small and large colonies are isolated and the authors show that the large colonies have undergone fusion of chromosome 10 with various other chromosomes. They also claim that the neocentromeres become inactivated upon chromosome fusion.

Although this manuscript is interesting, it has a preliminary feel. Whilst it is interesting that neocentromeres can form in Cryptococcus, which, like Candida, is a human pathogen, it is questionable whether the study provides significant advances in understanding the processes of neocentromere formation. The manuscript would also benefit from significant rewriting.

There is little discussion of potential relevance to the virulence / pathogenicity of *Cryptococcus deuterogattii*.

Several points need to be addressed before the manuscript could be considered for publication in *eLife*.

A more detailed description of *C. deuterogattii* centromeres is required – size, how many types of transposons are present? What is the level of sharing between centromeres? How much unique sequence is at each centromere? In particular a more-detailed diagram of cen10 is needed.

How many total transformants were obtained from the 99 transformations? Was it only the 7 mentioned? Or were there many NAT/NEO positive transformants that were incorrect? The approach used relies on correct integration and neocentromere formation to both occur almost in quick succession in order to recover the desired isolates. Did the authors consider/try a “two-step” approach such as that used in *S. pombe* (Ishii et al., 2008), in which a split marker gene is united via Cre-lox to delete the centromere?

Three strains are analysed by CENPA ChIP-seq and two neocentromeres are identified. *Cen10Δ*-1 has CENPA enrichment at both neocen1 and neocen2 sites, *cen10Δ*-2 has only neocen1, and *cen10Δ*-3 has both. What about the other neocen strains (*cen10Δ*-4,5,6,7)? Where is CENPA located in these other isolates, especially in *cen10Δ*-5?

Figure 1: cen10 and the precise region deleted must be indicated (either as a bar above the region or with shading that extends down Figure 1A). Relevant features, such as the transposon remnants at cen10, neocen1 (or A), neocen2 (or B), the gene names for the genes that are mentioned in the text and/or analysed in other figures must be labelled.

I assume that the reads mapping to the (deleted) cen10 region in *cen10Δ*-1,-2,-3 are due to CENPA ChIP-seq reads from copies of the transposons at other centromeres mapping to cen10 in the reference genome? This must be explained in the legend.

In part B the peak height at *cen10Δ*-2 neocen1 looks higher than those in *cen10Δ*-1, and -3. But in C – a zoom-in of that region – they all look similar. Also the “background”? across the chromosome arm in *cen10Δ*-1 appears higher than in the other isolates – what is real and what is due to differences in what is being displayed / scales?

The information associated with all ChIP-seq data (and similar) figures must be improved. The y axes are not labelled. What exactly is plotted? Has the data been normalized to input? Are the scales the same in all parts of the figure (the absence of any numbers on the y axes makes it impossible to interpret)? The legends must be more precise and informative.

What are the segregation properties of chr10 in the *cen10Δ* strains? Do neocen1 and/or neocen2 behave as centromeres? Ideally this should be done by integration of lacO arrays on chr10 and visualization via LacI-GFP. Integration of arrays near neocen1 and (separately) near cen10/neocen2 would enable predictions about the behavior of different loci to be tested. For instance, one would expect the neocen2 locus to be far from the spindle pole body in wild type (non-neocen) cells but close to it in *cen10Δ*-2/neocen cells. In addition, a single chr10 bearing two neocentromeres might exhibit segregation defects such as lagging/stretched chromosomes or chromosome breakage. Introduction of lacO arrays should be attempted, alternatively *cen10Δ* cells should at least be stained for CENPA and DAPI and cells with observable chromosome segregation defects quantified.

To assess whether either/both neocentromere is assembling a bona fide kinetochore ChIP-qPCR should be performed for a kinetochore protein (ideally an outer kinetochore protein). (Subsection “Deletion of centromere 10 results in neocentromere formation” paragraph three)

Differences in colony size are mentioned multiple times in the manuscript. It would be of interest to see examples of colonies of the various *cen10Δ* strains at different temperatures (in addition to the growth curves).

Figure 2: Has the qRT-PCR data presented in Figure 2 been normalized in any way to account for the ploidy difference between the wild-type strain and the *cen10Δ* strains? For instance, *cen10Δ*-2-S3 (which is presumably similar to *cen10Δ*-2) has a chr10 ploidy of 1.4 X wild-type. qPCR could be performed on genomic DNA (relative to a control euploid locus) and the qRT-PCR expression data normalized to it. If such normalization has been done already it is not described in the manuscript (and it should be).

Figure 4: Is it known where on chr10 the extra centromere-containing region of chromosome 5 is in *cen10Δ*-5-S3? It is intriguing that the neocen1 region appears to be duplicated in *cen10Δ*-L2, and, based on PFGE it is a duplication on chr10. What are the levels of CENPA on (duplicated?) neocen1? Is a double copy of neocen1 giving improved chromosome segregation?

It is interesting that cen10-Δ5 derivatives are euploid for chr10 but aneuploid for chr8. What might be the explanation for this observation? Are two copies of chr8 somehow beneficial?

“In contrast to the other fused chromosomes, this chromosomal fragment did not fuse to a chromosome with a native centromere, and thus the mutant still had a fitness defect.”, this statement is too strong – it's speculation. “thus” should be replaced by “consistent with”.

Subsection “*cen10Δ* isolates are aneuploid”: *cen10Δ*-5 strains are not euploid for chromosome 8.

Figure 4 legend – indicates that endogenous cen10 reads are absent due to its deletion. However, there are reads in the ChIP-seq in Figure 1. What is the reason for this difference?

The claim in the Abstract that the chr10 neocentromere is inactivated upon chromosome fusion must be supported by more substantial data. Figure 6 needs major improvement. It shows CENPA ChIP-qPCR for only wt, *cen10Δ*-1L and *cen10Δ*-2L, analyzing neocen1 and neocen2 regions. The CENPA ChIP-qPCR for positive control ie cen10-Δ1 and cen10-Δ2 with “active” neocentromeres (neocen1 and neocen2) is missing. Without that it is hard to interpret the data.

Which large colonies are analysed? Are they the same ones as in Figure 4, 5? There should be an indication of which strain they correspond to, or they should be given a new number.

CENPA ChIP-qPCR should be performed on three biological replicates and graphs show standard deviation.

CENPA ChIP-qPCR should be done for all original strains (and/or S strains) and fast-growing (L) derivatives. Ideally including qPCR for cen10 (as appropriate) and the centromere of the fusion chromosomes.

It is interesting that CENPA is apparently present on rrd2-1/neocen2 in wild type cells (in qPCR at least). There is no comment on this in the text. Could it be that wild-type cells do contain a small amount of CENPA at this location and, in the case of centromere deletion, this seeds further CENPA deposition to establish a neocentromere?

Figure 6B and subsection “*cen10Δ* chromosome is rescued by chromosome fusion”. The evidence that *cen10Δ*-3-L1 has a fusion between chr10 and chr13 should be explicitly described. Has PCR been done to confirm this?

All *cen10Δ*-5 strains analysed by PFGE are aneuploid for chromosome 8. In the figure legend there is the speculation that this occurred during or before transformation. Such speculative statements should not be placed in figure legends. I understand from the text and strain list that strain *cen10Δ*-5 (KS6) in the originally-derived and restreaked neocentromere strain and that the small and large colonies (KS21-25) are derived from that one original strain by growing at 37 and selecting small and large colonies. So the chr8 aneuploidy could have arisen in KS6 after transformation.

Figure 6B. Conclusions from data must be stated in the text, not the legend! *Cen10Δ*-5-L1 – from the data presented in the figure it can only be concluded that the chromosome 10 band is gone, it could have fused with another chromosome resulting in a fusion chromosome that migrates at the same size as other chromosomes. Southern would need to be performed with a chr10 probe to rigorously assess this. The sequencing data also confirms the fusions. But what is the data for 10-13 fusion in *cen10Δ*-L2?

Figure legends should only describe what was done and what is presented so that the reader can understand the data. Legends should not contain interpretation or discussion. All legends should be reviewed and modified accordingly. In addition, the figures themselves would be far easier for the reader to comprehend if the labelling of figures was improved.

[Editors’ note: further revisions were suggested prior to acceptance, as described below.]

Thank you for resubmitting your work entitled "Centromere deletion in *Cryptococcus deuterogattii* leads to neocentromere formation and chromosome fusions" for further consideration by *eLife*. Your revised article has been evaluated by Kevin Struhl (Senior Editor) and Wolf Heyer (Reviewing Editor) as well as two additional reviewers.

The manuscript has substantially improved and all reviewers recognized the strong efforts put into this revision. There are some remaining issues that need to be addressed before acceptance, as outlined below:

1) The remaining revisions do not require new experimentation.

2) The major point is the analysis of the ChIP seq experiments, which should change to fold enrichment over input on the y axis in ChIP-seq analyses.

Original comment:

In part B the peak height at *cen10Δ*-2 neocen1 looks higher than those in *cen10Δ*-1, and -3. But in C – a zoom-in of that region – they all look similar. Also, the “background”? across the chromosome arm in *cen10Δ*-1 appears higher than in the other isolates – what is real and what is due to differences in what is being displayed / scales?

Author response:

We thank the reviewers for noticing these issues, which are due to the "auto scale setting" feature of the IGV viewer. For the revised Figure 1, we set the read coverage to the maximum peak height, which reduced the background.

Reviewer comment:

OK, but it is more informative and best practice to indicate the fold enrichment over input on the y axis in ChIP-seq analyses.

Original comment:

The information associated with all ChIP-seq data (and similar) figures must be improved. The y axes are not labelled. What exactly is plotted? Has the data been normalized to input? Are the scales the same in all parts of the figure (the absence of any numbers on the y axes makes it impossible to interpret)? The legends must be more precise and informative.

Author resposne:

We have introduced "The read coverage (y-axis) shows the enrichment of CENP-A and

the x-axis shows the chromosome coordinates." All data shown are the result of the ChIP sample normalized with input. As expected, there is some variation in the number of reads of each ChIP analysis and this might be due to the ChIP, sequencing, library preparation, or experimental variation. For each sample, the reads are normalized to 1 within the sample.

Reviewer comment:

OK, but as above it is more informative and best practice to indicate the fold enrichment over input on the y axis in ChIP-seq analyses

3) The quality of the graphs, e.g. those presented in Figures 2 and 6 could be improved.

4) What is the strange effect on the ethidium-stained gels in Figure 1—figure supplement 1 and Figure 5—figure supplement 1?

5) The Discussion is quite lengthy and would benefit from being edited down. There are many of redundant descriptions. It is not necessary to summarize results there.

6) “*C. deuterogattii* is responsible for an ongoing outbreak in the Pacific Northwest regions of Canada and the United States” – outbreak of what?

7) In some places in the text it would be helpful to refer to specific parts of figures, e.g. Figure 3C.

8) Please indicate the precise region deleted in *cen9Δ* and *cen10Δ* in Figure 1. Discuss *cen9Δ* and *cen10Δ* in same order in text and figure.

---

## [Author Response]

[Editors’ note: the authors resubmitted a revised version of the paper for consideration. What follows is the authors’ response to the first round of review.]

Reviewer #1:This is a well written manuscript analyzing the consequences of centromere 10 deletion in the fungal pathogen Cryptococcus deuterogattii. The authors identify the formation on neocentromeres at two different locations on chromosome 10. Elevated growth temperature selects for chromosome fusions of chromosome 10 with another chromosome involving inactivation of the neocentromere. The events described in this work present a mechanism that could explain the changes in genome organization within the Cryptococcus pathogenic species complex.The evidence presented supports the conclusions, and my main comment is that the analysis of the 2 isolates with 2 mapped neocentromeres remains inconclusive.Essential revision:1) The authors identify 2 isolates that contain 2 neocentromeres on chromosome 10 as defined by CENP-A ChIP. They present 4 models to explain the observation of having 2 CENP-A enrichment loci in a single chromosome: 1) aneuploidy, 2) a dicentric, 3) instability and neo-centromere switching, 4) only one CENPA locus being a functional centromere. The authors show that only a minority fraction of cells is aneuploid based on sequence read analysis. In the Discussion, they argue against a switching model, and discuss the possibility of the other two model. This is not very satisfying, and a ChIP experiment using a kinetochore component could be conducted to assess the functionality of both CENPA loci to assemble a kinetochore. There may be alternative routes to resolve this.

Thank you for summarizing the findings, models, and sequencing. To confirm the CENP-A ChIP data, we have transformed a large subset of the centromere mutants with genes encoding either CENP-C or Mis12 epitope-tagged with GFP. As *cen10*Δ mutant-A, -B and -C span the gene encoding CENP-C we have transformed these isolates with Mis12-GFP. The *cen9*Δ mutants were transformed with CENP-C-GFP. Subsequently, we performed ChIP-qPCRs and this experiment showed that Mis12 and CENP-C co-localize with the CENP-A-enriched regions and confirmed that the neocentromeres are functional centromeres. The fact that we still see two enriched CENP-A regions in *cen*Δ-A and -C suggests that the observation is real; however, we cannot exclude the models above. Unfortunately, we cannot perform single-cell ChIPs and with the current experiment we are sequencing a population of cells. For example, we noticed when we streak purify *cen10*Δ cells, we obtain a population of colonies of mixed sizes (large and small). When we streak purify the small colonies, we again obtain a population of mixed size colonies, which is in contrast to streak purifying large colonies which only subsequently produce large colonies. As we show in the manuscript, the large colonies are the products s of chromosome fusion and the neocentromere is silenced.

In addition, we have identified the neocentromeres of *cen10*Δ-D to -G and have six mutants wherein centromere 9 is deleted. This allowed us to compare the neocentromeres on different chromosomes.

Reviewer #2:Cryptococcus deuterogattii has regional centromeres and lacks active transposons. In this paper, authors deleted centromere 10 and examined chromosomes in surviving clones after centromere deletion (Cen10Δ mutants). This deletion caused neocentromere formation in two particular sites. Unlike neocentromere observed in other organisms, neocentromere formation in C. deuterogattii did not prevent expression of gene that cover neocentromeres. Cen10Δ mutants showed growth defects and aneuploidy. If growth temperature was increased, neocentromeres were inactivated and chromosome-fusion occurred. Finally authors speculated that these observations might occur under natural situation during speciation.This is potentially interesting study, but work in this study is relatively descriptive. My major concern is unclearness how authors' observation in this study will be generalized. Authors performed various nice experiments and data are clear, however, I am not sure whether chromosomes with a neocentromere always get fusion with other chromosomes with growth fitness. In addition, it is also unclear whether cells with a neocentromere always show aneuploid. I agree that neocentromeres in this study are relatively weak compared with native centromeres, but I am not sure whether neocentromeres in C. deuterogattii are always weak or not. If authors show an evidence that observations in this study frequently occur or they can clearly explain about generality of this work, my mind will be changed.

Thank you for your comments. To answer the question raised, we have deleted another centromere (*CEN9*) and have identified the neocentromeres of *cen10*Δ isolates -D to -G. Only *cen10*Δ mutants have reduced fitness and undergo chromosome fusion. Centromere 9 mutants also exhibit a slow-growth phenotype, but the fitness defect is not as pronounced as in the *cen10*Δ mutants and we did not observe a population of mixed colony sizes in *cen9*Δ mutants. In the revised manuscript evidence is presented that the *cen10*Δ neocentromeres may be less fit than the native centromere and at higher growth temperature chromosome fusion has occurred in isolates producing large colonies. This shows a parallel with previous studies on neocentromere formation in *S. pombe.* However, in *S. pombe,* upon centromere deletion, either neocentromere formation or chromosome fusion occurs. Based on de novogenome assemblies, three chromosome fusions were identified and confirmed by PFGE, indicating that chromosome fusion occurred in all of the large *cen10*Δ isolates analyzed. In addition, the neocentromeres of *cen9*Δ and *cen10*Δ span actively transcribed genes, which has not been shown before. In *S. pombe* and *C. albicans* the genes are silenced. In chickens, genes are expressed in neocentromeres; however, these chromosomes are diploid and the neocentromere could bind to one only chromosome and the gene on the other chromosome could still be expressed.

Reviewer #3:Centromeres vary in size and sequence between organisms. In most organisms studied, the centromere is marked by the presence of the histone H3 variant, CENPA. Accumulated evidence indicates that CENPA is the epigenetic mark that specifies centromere identity. Consistent with an epigenetic mode of centromere propagation in which sequence does not dominate, neocentromeres can arise on chromosomal regions which have no homology to centromere sequences. Many naturally-occurring neocentromeres have been reported and neocentromeres have been experimentally induced in several organisms, including fission yeast, Candida and chicken.This manuscript describes the isolation and characterization of neocentromere strains in the human pathogen Cryptococcus deuterogattii, using an approach similar to previous studies in other species including Candida albicans. The authors delete the centromere of chromosome 10 and recover rare survivors which they go on to show by CENPA ChIP-seq have acquired either one or two neocentromeres on chromosome 10. They report that this has little impact on underlying gene expression. By whole genome sequencing the authors provide evidence that the strains have a degree of aneuploidy, and are growth impaired. By incubation of neocentromere strains at 37^o^C, small and large colonies are isolated and the authors show that the large colonies have undergone fusion of chromosome 10 with various other chromosomes. They also claim that the neocentromeres become inactivated upon chromosome fusion.Although this manuscript is of interest, it has a preliminary feel. Whilst it is interesting that neocentromeres can form in Cryptococcus, which, like Candida, is a human pathogen, it is questionable whether the study provides significant advances in understanding the processes of neocentromere formation. The manuscript would also benefit from significant rewriting.There is little discussion of potential relevance to the virulence / pathogenicity of Cryptococcus deuterogattii.

Thank you for the comments and suggestions. We have included a section about possible effects on virulence in the Discussion. The competition assays showed that both *cen9*Δ and *cen10*Δ mutants grow slower than the wild type. If centromere deletions occurred in nature, we hypothesize that the wild type would outcompete *cen*Δ isolates. Based on the reduced fitness of the *cen*Δ mutants we hypothesize that *cen*Δ mutants would be less virulent than the wild type.

Several points need to be addressed before the manuscript could be considered for publication in eLife.A more detailed description of C. deuterogattii centromeres is required – size, how many types of transposons are present? What is the level of sharing between centromeres? How much unique sequence is at each centromere? In particular a more-detailed diagram of cen10 is needed.

The centromeres of *C. deuterogattii* were described in detail in a recent publication of the lab (Yadav et al., 2018). In that article the centromere content was analyzed in great detail, including the transposable element composition of the centromeres.

In the Introduction we have added more information about the size of *C. deuterogattii* and *C. neoformans* centromeres. “Recently, the centromeres of the human pathogenic fungus *Cryptococcus deuterogattii* were characterized and compared to those of the closely related species *Cryptococcus neoformans* (centromeres ranging from 27 to 64 kb), revealing dramatically smaller centromeres in *C. deuterogattii* (ranging from 8.7 to 21 kb) (Janbon et al., 2014; Yadav et al., 2018).” In the updated figures of the revised manuscript, we have employed the same color scheme for Tcn1-6 as presented by Yadav et al., 2018.

How many total transformants were obtained from the 99 transformations? Was it only the 7 mentioned? Or were there many NAT/NEO positive transformants that were incorrect? The approach used relies on correct integration and neocentromere formation to both occur almost in quick succession in order to recover the desired isolates. Did the authors consider/try a “two-step” approach such as that used in *S. pombe* (Ishii et al., 2008), in which a split marker gene is united via Cre-lox to delete the centromere?

We only saw 14 false positives for all 99 biolistic transformations. For the false positives, the *NAT*/*NEO* cassette was integrated randomly in the genome and these transformants were not characterized further. Thus, a total of 21 transformants were obtained, of which 7 were bona fide *cen10*Δ mutants (a frequency of 33%). While deleting genes for other projects, we have observed that the homologous recombination transformation rate is high in *C. deuterogattii*.

Thank you for the suggestion to use the Cre-lox method for the deletion of centromeres. To our knowledge, this method has not been developed or applied for *C. deuterogattii*. Instead, we have applied a CRISPR-Cas9 based homologous recombination method to delete centromere 9. This resulted in the successful deletion of centromere 9. The efficiency of centromere 9 deletion was significantly improved; however, the false positive (*NAT* ectopic integration in the genome) rate was significantly higher as well. As we have used CRISPR-Cas9, we did not make any conclusions for the transformation rate to delete centromere 9.

Three strains are analysed by CENPA ChIP-seq and two neocentromeres are identified. Cen10Δ-1 has CENPA enrichment at both neocen1 and neocen2 sites, cen10Δ-2 has only neocen1, and cen10Δ-3 has both. What about the other neocen strains (cen10Δ-4,5,6,7)? Where is CENPA located in these other isolates, especially in cen10Δ-5?

We have performed ChIP-seq for all of the *cen10*Δ mutants (-A to -G) and included the neocentromeres of *cen10*Δ -D to -G in our analysis in the revised manuscript. In addition, we obtained six *cen9*Δ mutants and have performed ChIP-seq for all of these mutants as well. This allowed us to define the chromosomal position for all of the neocentromeres. The neocentromere of *cen10*Δ-E is located directly adjacent to the left telomere. In the manuscript we speculate that this might be a reason why the centromere deletion mutant does not exhibit a fitness defect compared to the other *cen10*Δ mutants.

Figure 1: cen10 and the precise region deleted must be indicated (either as a bar above the region or with shading that extends down Figure 1A). Relevant features, such as the transposon remnants at cen10, neocen1 (or A), neocen2 (or B), the gene names for the genes that are mentioned in the text and/or analysed in other figures must be labelled.

*C. deuterogattii* has only truncated transposable elements and these elements are only present in the native centromeres. We have indicated the truncated transposable elements present in the native centromere (*CEN9* and *CEN10*) with colors and labeled Tcn4 (orange) and Tcn6 (green). The neocentromeres are formed in unique chromosomal locations that lack transposable elements and repeats. Also, the neocentromeres span genes and are flanked by genes. We have updated Figure 1 and have labeled all of the genes.

I assume that the reads mapping to the (deleted) cen10 region in cen10Δ-1,-2,-3 are due to CENPA ChIP-seq reads from copies of the transposons at other centromeres mapping to cen10 in the reference genome? This must be explained in the legend.

Yes, this is correct and we have added this information in the revised figure legend. For the same reason, we still see reads mapped back to the deleted native centromere in *cen9*Δ-D. This is absent in other *cen9*Δ mutants; one possibility is that this is due to the ChIP library preparation and some reads are more enriched than others.

In part B the peak height at cen10Δ-2 neocen1 looks higher than those in cen10Δ-1, and -3. But in C – a zoom-in of that region – they all look similar. Also the “background”? across the chromosome arm in cen10Δ-1 appears higher than in the other isolates – what is real and what is due to differences in what is being displayed / scales?

We thank the reviewers for noticing these issues, which are due to the “auto scale setting” feature of the IGV viewer. For the revised Figure 1, we set the read coverage to the maximum peak height, which reduced the background.

The information associated with all ChIP-seq data (and similar) figures must be improved. The y axes are not labelled. What exactly is plotted? Has the data been normalized to input? Are the scales the same in all parts of the figure (the absence of any numbers on the y axes makes it impossible to interpret)? The legends must be more precise and informative.

We have introduced “The read coverage (y-axis) shows the enrichment of CENP-A and the x-axis shows the chromosome coordinates.” All data shown are the result of the ChIP sample normalized with input. As expected, there is some variation in the number of reads of each ChIP analysis and this might be due to the ChIP, sequencing, library preparation, or experimental variation. For each sample, the reads are normalized to 1 within the sample.

What are the segregation properties of chr10 in the cen10Δ strains? Do neocen1 and/or neocen2 behave as centromeres? Ideally this should be done by integration of lacO arrays on chr10 and visualization via LacI-GFP. Integration of arrays near neocen1 and (separately) near cen10/neocen2 would enable predictions about the behavior of different loci to be tested. For instance, one would expect the neocen2 locus to be far from the spindle pole body in wild type (non-neocen) cells but close to it in cen10Δ-2/neocen cells. In addition, a single chr10 bearing two neocentromeres might exhibit segregation defects such as lagging/stretched chromosomes or chromosome breakage. Introduction of lacO arrays should be attempted, alternatively cen10Δ cells should at least be stained for CENPA and DAPI and cells with observable chromosome segregation defects quantified.

Thank you for the suggestion to test for segregation properties. We have tried several times to cross the *cen10*Δ mutants. However, even for the wild-type controls it is challenging to observe mating for *C. deuterogattii*. Unfortunately, due to this technical limitation at present the ability to test the segregation properties of the neocentromeres is limited. To verify if the neocentromeres are functional ChIP-qPCR with Mis12 or CENP-C were conducted, and enrichment for all earlier identified CENP-A enriched regions was observed. Studying the chromosomes of *C. deuterogattii* by microscopy has not as yet been accomplished or reported and is beyond the state of the art. The chromosomes are small and only one mCherry/GFP signal in the nucleus is visible.

To assess whether either/both neocentromere is assembling a bona fide kinetochore ChIP-qPCR should be performed for a kinetochore protein (ideally an outer kinetochore protein). (Subsection “Deletion of centromere 10 results in neocentromere formation” paragraph three)

We have transformed a large subset of neocentromere mutants with a gene encoding an additional epitope-tagged kinetochore protein. We have analyzed *cen10*Δ mutants with Mis12- GFP and *cen9*Δ mutants with CENP-C-GFP. Subsequently ChIP-qPCR was performed. The results obtained confirm the neocentromeric (based on CENP-A enrichment) location in *cen*Δ mutants.

Differences in colony size are mentioned multiple times in the manuscript. It would be of interest to see examples of colonies of the various cen10Δ strains at different temperatures (in addition to the growth curves).

We have included a supplementary figure with an example of a population of colonies with a mixed colony size for a centromere 10 deletion mutant.

Figure 2: Has the qRT-PCR data presented in Figure 2 been normalized in any way to account for the ploidy difference between the wild-type strain and the cen10Δ strains? For instance, cen10Δ-2-S3 (which is presumably similar to cen10Δ-2) has a chr10 ploidy of 1.4 X wild-type. qPCR could be performed on genomic DNA (relative to a control euploid locus) and the qRT-PCR expression data normalized to it. If such normalization has been done already it is not described in the manuscript (and it should be).

Thank you for this suggestion. For the revised Figure 2, we have normalized gene expression of the *cen10*Δ mutants with a housekeeping gene located on chromosome 10 and normalized to the wild-type. All *cen9*Δ mutants are normalized to the wild type and actin.

Figure 4: Is it known where on chr10 the extra centromere-containing region of chromosome 5 is in cen10Δ-5-S3? It is intriguing that the neocen1 region appears to be duplicated in cen10Δ-L2, and, based on PFGE it is a duplication on chr10. What are the levels of CENPA on (duplicated?) neocen1? Is a double copy of neocen1 giving improved chromosome segregation?

ChIP-seq has not been conducted with these isolates and the centromeric region of the fused chromosomes is not at present established

It is interesting that cen10-Δ5 derivatives are euploid for chr10 but aneuploid for chr8. What might be the explanation for this observation? Are two copies of chr8 somehow beneficial?

Thank you for the suggestion; we are not aware that aneuploidy for chromosome 8 can be beneficial. However, this is an interesting observation.

“In contrast to the other fused chromosomes, this chromosomal fragment did not fuse to a chromosome with a native centromere, and thus the mutant still had a fitness defect.”, this statement is too strong – it's speculation. “thus” should be replaced by “consistent with”.

Thank you for the suggestion, we have replaced “thus” with “consistent with”

Subsection “cen10Δ isolates are aneuploid”: cen10Δ-5 strains are not euploid for chromosome 8.

We have re-written the sentence.

Figure 4 legend indicates that endogenous cen10 reads are absent due to its deletion. However, there are reads in the ChIP-seq in Figure 1. What is the reason for this difference?

This is due to the presence of repeats in the remaining native centromeres. The native centromeres span truncated transposable elements and repeats. Due to the mapping with Bowtie2, the reads are mapped equally between repeats. All of the data shown is based on mapping the reads to the wild-type genome assembly. Due to duplicated regions (TEs/repeats), sequencing reads derived from regions other than centromere 9 or 10 are mapped back to the location of the native centromere in the centromere deletion mutants.

We have confirmed the centromere mutants based on Southern blot analysis and PCR confirmation, which both document the lack of the native centromere in the centromere 9 or 10 mutants.

The claim in the Abstract that the chr10 neocentromere is inactivated upon chromosome fusion must be supported by more substantial data. Figure 6 needs major improvement. It shows CENPA ChIP-qPCR for only wt, cen10Δ-1L and cen10Δ-2L, analyzing neocen1 and neocen2 regions. The CENPA ChIP-qPCR for positive control ie cen10-Δ1 and cen10-Δ2 with “active” neocentromeres (neocen1 and neocen2) is missing. Without that it is hard to interpret the data.Which large colonies are analysed? Are they the same ones as in Figure 4, 5? There should be an indication of which strain they correspond to, or they should be given a new number.CENPA ChIP-qPCR should be performed on three biological replicates and graphs show standard deviation.

Thank you for the comment. All qPCR results shown in the manuscript were performed in three replicates and the standard deviation for each measurement has been included in the revised manuscript. The controls suggested (*cen10*Δ-A and *cen10*Δ-B) have also now been included. For *cen10*Δ-A and *cen10*Δ-A-L1 we have indicated the two CENP-A-enriched regions. For *cen10*Δ-B and *cen10*Δ-B-L2 we have used the same primer set, although this *cen10*Δ mutant only has one CENP-A-enriched region. As control, the wild type is presented. The same strains in Figure 4 and 5 were used for this analysis and we have added this information in the figure legend of Figure 6 (*cen10*Δ-A-L1 and *cen10*Δ-B-L2).

CENPA ChIP-qPCR should be done for all original strains (and/or S strains) and fast-growing (L) derivatives. Ideally including qPCR for cen10 (as appropriate) and the centromere of the fusion chromosomes.

We now have included qPCR data for an additional kinetochore protein and this confirms the CENP-A sequencing shown in Figure 1. This is based on independent experiments and before the ChIPs were performed, we had to transform all of the original *cen*Δ mutants. After transformation, all of the transformants obtained were streak purified twice.

When we streak purify *cen10*Δ mutants for single colonies, we always observe a mixed population of colony sizes. When we streak purify small *cen10*Δ colonies we again always see a mixed population. Streak purification of large *cen10*Δ colonies instead results in only large colonies. The growth curves of Figure 3—figure supplement 1 shows that small colonies grow like the original mutants and, in contrast. the large colonies have wild-type fitness levels. All of the ChIP experiments are based on large volumes as we cannot conduct single cell ChIPs. As mentioned, we cannot perform ChIP-qPCRs for the native centromere 10, as this centromere has been deleted.

Based on all of these experiments and conclusions we hypothesize that all small colonies of the same *cen10*Δ have the same neocentromeric location and all large colonies have inactive neocentromeres.

It is interesting that CENPA is apparently present on rrd2-1/neocen2 in wild type cells (in qPCR at least). There is no comment on this in the text. Could it be that wild-type cells do contain a small amount of CENPA at this location and, in the case of centromere deletion, this seeds further CENPA deposition to establish a neocentromere?

The qPCRs shown in Figure 6 show a slight enrichment of CENP-A on the *RRD2* gene. However, this was only in one region located in the gene, and the second region was not enriched. We now included CENP-A reads of the wild type and show this for all chromosomal location where neocentromeres are formed. Based on ChIP-seq we do not see any enrichment for CENP-A in the *RRD2*-2 region.

Figure 6B and subsection “cen10Δ chromosome is rescued by chromosome fusion”. The evidence that cen10Δ-3-L1 has a fusion between chr10 and chr13 should be explicitly described. Has PCR been done to confirm this?

Long-read sequencing data for this strain is not available and thus it is challenging to design a primer pair to validate the fusion junction. However, all of the chromosome fusions based on de novogenome assembly were confirmed by PCR and this information is now included as a panel in Figure 5—figure supplement 1. T. Beside the PCR confirmation, the chromosome fusions were confirmed by read coverage. In the de novoassembly, the reads show a continuous pattern and there are no read breaks.

All cen10Δ-5 strains analysed by PFGE are aneuploid for chromosome 8. In the figure legend there is the speculation that this occurred during or before transformation. Such speculative statements should not be placed in figure legends. I understand from the text and strain list that strain cen10Δ-5 (KS6) in the originally-derived and restreaked neocentromere strain and that the small and large colonies (KS21-25) are derived from that one original strain by growing at 37 and selecting small and large colonies. So the chr8 aneuploidy could have arisen in KS6 after transformation.

Thank you for this suggestion; we have removed this speculation. We can only speculate when the chr8 aneuploidy occurred in this mutant. The chromosome duplication might have occurred after the transformation event. As we see this chromosome duplication for all small and large colonies it suggests that all of the cells have the parental chr8 duplication.

Figure 6B. Conclusions from data must be stated in the text, not the legend! Cen10Δ-5-L1 – from the data presented in the figure it can only be concluded that the chromosome 10 band is gone, it could have fused with another chromosome resulting in a fusion chromosome that migrates at the same size as other chromosomes. Southern would need to be performed with a chr10 probe to rigorously assess this. The sequencing data also confirms the fusions. But what is the data for 10-13 fusion in cen10Δ-L2?

We did not include this strain in our sequencing experiment.

Figure legends should only describe what was done and what is presented so that the reader can understand the data. Legends should not contain interpretation or discussion. All legends should be reviewed and modified accordingly. In addition, the figures themselves would be far easier for the reader to comprehend if the labelling of figures was improved.

We have now updated all figure legends to enhance the labeling and presentation of the content.

[Editors’ note: what follows is the authors’ response to the second round of review.]

The manuscript has substantially improved and all reviewers recognized the strong efforts put into this revision. There are some remaining issues that need to be addressed before acceptance, as outlined below:1) The remaining revisions do not require new experimentation.2) The major point is the analysis of the ChIP seq experiments, which should change to fold enrichment over input on the y axis in ChIP-seq analyses.Original comment:In part B the peak height at cen10Δ-2 neocen1 looks higher than those in cen10Δ-1, and -3. But in C – a zoom-in of that region – they all look similar. Also, the “background”? across the chromosome arm in cen10Δ-1 appears higher than in the other isolates – what is real and what is due to differences in what is being displayed / scales?Author response:We thank the reviewers for noticing these issues, which are due to the "auto scale setting" feature of the IGV viewer. For the revised Figure 1, we set the read coverage to the maximum peak height, which reduced the background.Reviewer comment:OK, but it is more informative and best practice to indicate the fold enrichment over input on the y axis in ChIP-seq analyses.Original comment:The information associated with all ChIP-seq data (and similar) figures must be improved. The y axes are not labelled. What exactly is plotted? Has the data been normalized to input? Are the scales the same in all parts of the figure (the absence of any numbers on the y axes makes it impossible to interpret)? The legends must be more precise and informative.Author resposne:We have introduced "The read coverage (y-axis) shows the enrichment of CENP-A andthe x-axis shows the chromosome coordinates." All data shown are the result of the ChIP sample normalized with input. As expected, there is some variation in the number of reads of each ChIP analysis and this might be due to the ChIP, sequencing, library preparation, or experimental variation. For each sample, the reads are normalized to 1 within the sample.Reviewer comment:OK, but as above it is more informative and best practice to indicate the fold enrichment over input on the y axis in ChIP-seq analyses.

Thank you for this comment. We have now indicated the height of the CENP-A peaks on the right side of each panel in the revised version of Figure 1. The height of the ChIP-seq peak shows the fold enrichment of the CENP-A peak when subtracted from the input sample.

3) The quality of the graphs, e.g. those presented in Figures 2 and 6 could be improved.

Thank you for this suggestion. To improve the figure quality we have removed the pattern of the bar plots and we use now a solid grey fill for the revised versions. In addition, each bar plot now has a black border and the error-bars are have been re-colored to black.

4) What is the strange effect on the ethidium-stained gels in Figure 1—figure supplement 1 and Figure 5—figure supplement 1?

Thank you for this comment; to be honest we don’t have a good explanation. We might have exceeded the voltage limits while running the gels as the shape of the ladder bands show a V-shape. Another option could be improper mixture of the agarose gel; however, this seems less unlikely as each PCR series was run on independent agarose gels.

5) The Discussion is quite lengthy and would benefit from being edited down. There are many of redundant descriptions. It is not necessary to summarize results there.

Thank you for this comment. As suggested, we have removed redundant text and the revised Discussion is now ~1.2 pages shorter. In addition, two short paragraphs were moved to the revised Results section.

6) “C. deuterogattii is responsible for an ongoing outbreak in the Pacific Northwest regions of Canada and the United States” – outbreak of what?

The outbreak referred to is that of cryptococcosis in the Pacific Northwest. The sentence in the revised manuscript has been replaced by: “*C. deuterogattii* is responsible for an ongoing cryptococcosis outbreak in the Pacific Northwest regions of Canada and the United States (Fraser et al., 2005).”

7) In some places in the text it would be helpful to refer to specific parts of figures, e.g. Figure 3C.

Thank you for this comment. We have inserted several references to the figures in the text and this should make the presentation more readily understood by the reader.

8) Please indicate the precise region deleted in cen9Δ and cen10Δ in Figure 1. Discuss cen9Δ and cen10Δ in same order in text and figure.

The deleted region of chromosome 9 in Figure 1A and the deleted region of chromosome 10 in Figure 1D are now indicated with a grey line in the revised figure. In the revised manuscript, each paragraph starts now with the description of results of centromere 9 followed by centromere 10.